# Beyond 2:4: Exploring V:N:M Sparsity for Efficient Transformer Inference on GPUs

## Abstract

To date, 2:4 sparsity has stood as the only sparse pattern that can be accelerated using sparse tensor cores on GPUs. In practice, 2:4 sparsity often possesses low actual speedups ($\leq 1.3$) and requires fixed sparse ratios, meaning that other ratios, such as 4:8, 8:16, or those exceeding 50% sparsity, do not incur any speedups on GPUs. Recent studies suggest that V:N:M sparsity is promising in addressing these limitations of 2:4 sparsity. This sparsity divides a weight matrix into multiple V×M blocks, pruning (M-4) columns within each block and applying 2:4 sparsity to the remaining columns. V:N:M sparsity inherently encompasses 2:4 sparsity but allows for higher and more flexible pruning ratios, typically resulting in greater practical speedups. However, regarding accuracy, the effects of V:N:M sparsity on broader Transformer models, such as vision Transformers and large language models (LLMs), are largely unexamined. Moreover, Some specific issues related to V:N:M sparsity, such as how to select appropriate V and M values, remain unresolved. In this study, we thoroughly investigate the application of V:N:M sparsity in vision models and LLMs across multiple tasks, from pretaining to downstream tasks. We propose three key approaches to enhance the applicability and accuracy of V:N:M-sparse Transformers, including heuristic V and M selection, V:N:M-specific channel permutation and three-staged LoRA training techniques. Experimental results show that, with our methods, the DeiT-small achieves lossless accuracy at 64:2:5 sparsity, while the DeiT-base maintains accuracy even at 64:2:8 sparsity. In addition, the fine-tuned LLama2-7B at 64:2:5 sparsity performs comparably or better than training-free 2:4 sparse alternatives on downstream tasks. More importantly, V:N:M-sparse Transformers offer a wider range of speedup-accuracy trade-offs compared to 2:4 sparsity. Overall, our exploration largely facilitates the V:N:M sparsity to act as a truly effective acceleration solution for Transformers in cost-sensitive inference scenarios.

## 1 Introduction

Transformer has gained significant popularity as backbones across various domains due to its remarkable performance in data modeling and scalability. However, Transformers are often characterized by a large number of parameters and high computational demands, resulting in prolonged inference latency. It is essential to compress Transformer models for efficient inference, especially in resource-constrained or latency-sensitive applications.

One possible way to accelerate Transformers is 2:4 sparsity, where only two out of every consecutive four parameters are retained in weight tensors. 2:4 sparsity is widely supported by Nvidia Ampere or newer GPUs. However, the current ecosystem for 2:4 sparsity exhibits three weaknesses that are rarely addressed. **1) Low practical speedups.** Unlike the theoretical claims of a twofold speedup, in most cases, neural networks with 2:4 sparsity achieve only a speedup in the range of 1.1 to 1.3x (Cai, 2023; Pool et al., 2021). **2) Only one sparsity pattern, i.e., 2:4, can be accelerated.** Other patterns at 50% sparsity, like 4:8 and 8:16, cannot yield any speedups on existing GPUs. **3) Failure to exploit higher sparsity ratio.** For some Transformer models with high weight redundancy, or in scenarios where inference overheads are more sensitive while model accuracy can be relatively relaxed, the optimal sparsity ratio can be larger than 50%, and 2:4 sparsity cannot fully leverage the potential performance gains from higher sparsity levels.

To address the weaknesses, Castro et al. (2023) proposes V:N:M sparsity. As shown in Figure 1, V:N:M sparsity divides the weight matrices of linear layers in Transformers into multiple blocks, each sized V × M. Within each block, (M - 4) columns are pruned, leaving 4 columns that implement 2:4 sparsity. V:N:M sparsity enables practical speedups for sparsity above 50% on GPUs. Notably, any GPU that supports 2:4 sparsity can also accelerate V:N:M sparsity. Due to higher compression ratios, V:N:M sparse Transformers deliver greater speedups compared to those using 2:4 sparsity.

The initial work on V:N:M sparsity primarily focuses on designing its acceleration kernel. Importantly, the impact of V:N:M sparsity on broader Transformer models, such as vision Transformers and large language models (LLMs) like the Llama series, remains under-explored. Additionally, fundamental issues regarding V:N:M sparsity, such as how to select appropriate values for V and M in a Transformer architecture, have never been resolved. Without addressing these issues, V:N:M sparsity can not comprehensively outperform 2:4 sparsity in compressing Transformers with high redundancy. In this work, we aim to bridge these gaps by systematically investigating the application of V:N:M sparsity across multiple Transformer models, with a particular emphasis on enhancing their accuracy. Specifically, our contributions are as follows:

- We propose a framework to enable the generation of highly accurate V:N:M-sparse Transformers under different constraints, which broadens the applicability of V:N:M sparsity.

- We propose three techniques to address challenges specific to V:N:M sparsity. First, we present a heuristic method for selecting V and M values that yield optimal accuracy-speedup trade-offs for V:N:M sparse Transformers. Second, we introduce V:N:M-specific channel permutation, which improves the accuracy of V:N:M-sparse Transformers within limited training budgets. Finally, we propose a three-stage LoRA training technique that adapts V:N:M sparsity for LLMs.

- Extensive experiments demonstrate the efficacy of our proposed scheme and techniques. Impressively, DeiT-base with 64:2:8 sparsity (75% sparse) achieves nearly lossless accuracy, with a minimal difference of less than 0.3% compared to the dense counterpart. As for speedups, the 64:2:8-sparse DeiT-base achieves a 1.7x speedup, while the 2:4 sparsity only provides a 1.15x speedup compared to the dense counterpart.

Our methods and results demonstrate that in scenarios with high inference costs or stringent latency requirements, V:N:M sparsity is a superior alternative to 2:4 sparsity for Transformers exhibiting significant redundancy. We advocate for the inclusion of V:N:M sparsity as a key consideration in deploying Transformers on GPUs that support 2:4 sparsity.

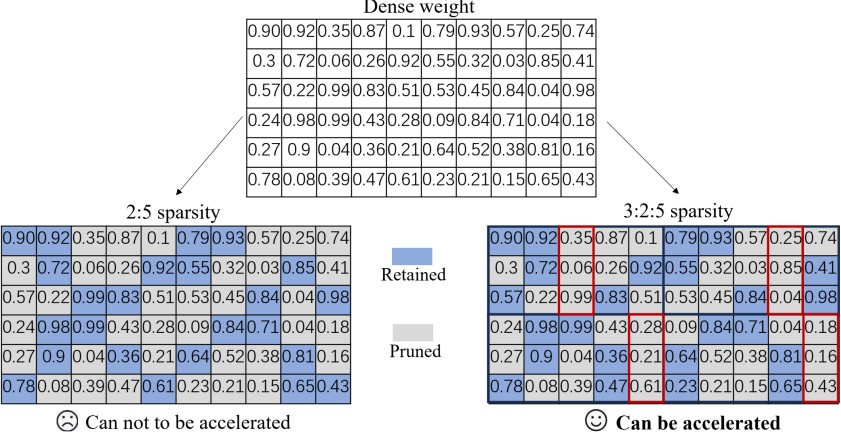

Figure 1: An example on N:M sparsity and V:N:M sparsity

## 2 RELATED WORK

**Sparsity in Transformers** Weight sparsity in Transformers can be categorized into three types: unstructured sparsity, like S$^2$ViTE (Chen et al., 2021b), SparseGPT (Frantar & Alistarh, 2023), and

Wanda (Sun et al., 2023); structured sparsity, represented by ViT-Slim (Chavan et al., 2022), VTP (Zhu et al., 2021), UVC (Yu et al., 2022), and SAViT (Zheng et al., 2022); and semi-structured sparsity. In particular, semi-structured sparsity generally offers a preferable trade-off between accuracy and speed. Yu et al. (2023) introduce 2:4 sparsity to vision Transformers, demonstrating that the DeiT series with 2:4 sparsity can maintain a nearly lossless performance. Xu et al. (2024) implement block-wise sparsity in Transformers, achieving notable speed improvements on neural processing units. Nevertheless, this approach results in unavoidable accuracy declines. In contrast, our V:N:M sparse DeiT-base can preserve nearly lossless accuracy even at 75% sparsity. Beyond weight sparsity, other components, such as tokens and attention heads, can also be pruned in Transformers. Some works in this aspect include T2T-ViT-24 (Yuan et al., 2021), PVT (Wang et al., 2021), Evo-ViT (Xu et al., 2022), EViT (Liang et al., 2022), DynamicViT (Rao et al., 2021), PS-ViT (Tang et al., 2022), and AdaViT (Yin et al., 2021). However, in this study, we primarily focus on the effects of V:N:M sparsity on Transformers, rather than extreme compressing a Transformer.

**Optimization for sparse Transformers** The sparse Transformers can be retrained to restore accuracy. The retraining process can be combined with techniques such as neural architecture search (Chen et al., 2021a; Chavan et al., 2022). During retraining, sparse masks can be updated periodically (Zhang et al., 2023; Lu et al., 2023). To address training instability, SR-STE (Zhou et al., 2021) proposes suppressing the weights that are masked out, allowing the updated masks to become progressively consistent as training advances. For large-scale Transformers, such as large language models (LLMs), post-training pruning (Frantar & Alistarh, 2023; Sun et al., 2023; Zhang et al., 2024) is effective when sparsity levels are low or moderate (less than 50%). Additionally, some studies (Kuznedelev et al., 2023; Kale-ab Tessera & Rosman, 2021) indicate that sparse networks are often under-trained provided with the same number of training epochs as their dense counterparts. Our study also finds that as training duration increases, the accuracy of sparse Transformers improves significantly, while dense Transformers exhibit minimal accuracy gains.

**Channel permutation** Channel permutation (CP) is extensively utilized in model quantization (Yuan et al., 2023) and sparsification (Ji et al., 2018; Tan et al., 2022; Lin et al., 2022; Pool & Yu, 2021). In particular for model sparsity, rearranging the order of weight or activation tensors prior to pruning can significantly reduce the subsequent one-shot pruning loss. Furthermore, as a specialized form of teleportation (Zhao et al., 2023; Mishkin et al., 2024), CP enhances the gradient flow of sparse models during training. However, other methods, such as extended training durations (Kuznedelev et al., 2023) and gradual pruning strategies (Bambhaniya et al., 2024; Jaiswal et al., 2022), can also improve gradient flow in sparse models, CP is particularly advantageous in low-training-budget scenarios, where CP can promote model convergence.

## 3 PRELIMINARY

**Pruning of V:N:M sparsity** Pruning a weight matrix $\mathbf{W}$ to achieve the V:N:M-sparse pattern involves three steps: 1) Calculate importance scores. First, compute the importance score for each weight in $\mathbf{W}$. 2) Column Pruning. Next, prune the columns within each V×M block. Within each block, the L1 norms of the importance scores for each column are compared, and the weights corresponding to minimal M−4 columns are pruned. 3) Conduct 2:4 Sparsity. After the column-wise pruning, each block retains exactly four columns. Subsequently, for each row, the weights corresponding to the last two importance scores are further pruned to establish the final V:N:M-sparse pattern. For descriptive convenience, we signify this V:N:N-sparse pruning process as $S_{V:N:M}$.

In this work, there are two commonly used criteria to form the importance score of a weight: the naive absolute values (ABS) and relative importance and activation (RIA) (Zhang et al., 2024). Specifically, RIA defines the importance score of a weight $\mathbf{W}_{ij}$ as:

$$RIA_{ij} = \left( \frac{|\mathbf{W}_{ij}|}{\sum |\mathbf{W}_{*j}|} + \frac{|\mathbf{W}_{ij}|}{\sum |\mathbf{W}_{i*}|} \right) \times \left( \|\mathbf{X}_i\|_2 \right)^a, \tag{1}$$

where $\sum |\mathbf{W}_{*j}|$ and $\sum |\mathbf{W}_{i*}|$ denote the summation of the absolute values of the input channel $j$ and output channel $i$, respectively. $\|\mathbf{X}_i\|_2$ is L2 norms of activations and $a$ is a factor to control the impact of activations on importance scores. Notably, both ABS and RIA are computationally efficient pruning criteria.

**Fixed and dynamic mask training for V:N:M sparsity** To restore the accuracy of V:N:M-sparse Transformers, sparse training is essential as V:N:M sparsity lies in high sparsity levels f at least 60% (V:2:5). At these high levels, merely applying post-training pruning is insufficient to reduce the significant accuracy loss. Specifically, after pruning a weight matrix, its 0-1 mask $\mathbf{M}$ that follows the V:N:M-sparse pattern can be easily derived. Denote the sparse weight matrix $\mathbf{W}' = \mathbf{W} \odot \mathbf{M}$, where $\odot$ is the element-wise multiplication operator. The weight update mechanism for fixed mask training is represented as:

$$\mathbf{W}'_t = \mathbf{W}'_{t-1} - \gamma \nabla_{\mathbf{W}'} \mathcal{L}_t(\mathbf{W}'_{t-1}) \tag{2}$$

Meanwhile, the weight update using dynamic mask training in the SR-STE framework is expressed as:

$$\mathbf{W}_t \leftarrow \mathbf{W}_{t-1} - \gamma \left( \nabla_{\mathbf{W}} \mathcal{L}_t(\mathbf{W}'_{t-1}) + \lambda \overline{\mathbf{M}}_{t-1} \odot \mathbf{W}_{t-1} \right) \tag{3}$$

In Eq. 2 and 3, $\nabla \mathcal{L}$ denotes the gradient, while $\gamma$ and $\lambda$ are the learning rate and regularization coefficient, respectively. $\overline{\mathbf{M}}_{t-1}$ represents the logical not operation of $\mathbf{M}_{t-1}$ at $t-1$, which enables regularization to only target the pruned weights and gradually decrease their norms. Eq. 2 indicates that only the retained weights are updated, with the V:N:M-sparse mask $\mathbf{M}$ remaining unchanged after one-shot pruning. In contrast, Eq. 3 gradient-updates the dense $\mathbf{W}$ and $\mathbf{M}$ is time-variant.

**Acceleration of V:N:M sparsity** As illustrated in Figure 2(b), the acceleration of V:N:M-sparse Transformers involves three steps: weight padding, conversion to compressed formats, and the application of V:N:M accelerated kernels. First, the weights of all linear layers are zero-padded to ensure that the input and output channels of weight matrices in linear layers are divisible by M and V, respectively. Next, the padded weights are converted into sparse storage formats that include only the compact non-zero values and their indices. The inference kernels then directly take these sparsely stored weights as input and leverage sparse tensor cores to accelerate V:N:M-sparse matrix multiplications (MMs) (Castro et al., 2023), as detailed in Appendix A. Due to the effective utilization of higher sparsity greater than 50%, V:N:M-sparse MMs possess fewer computations than 2:4-sparse MMs. Thus, for a dense Transformer, the V:N:M-sparse version typically achieves higher speedups than its 2:4 counterpart, with maximal end-to-end speedups reaching over 2x.

## 4 METHODS

**Scheme overview** We show the process of generating a V:N:M-sparse Transformer with high accuracy in Figure 2(a). Given a pretrained dense Transformer and a specified speedup threshold, a heuristic approach is employed to select appropriate V and M values for pruning the dense Transformer. After that, we consider two distinct scenarios. In the first scenario, where a limited training budget is available, V:N:M-specific CP, RIA-based pruning, and fixed mask training are sequentially employed. CP and RIA can significantly improve the accuracy of V:N:M-sparse Transformers upon pruning, while for low training budget constraints, fixed mask training, no matter with full-parameters or LoRA, incurs significantly lower overhead compared to dynamic mask training as the mask update costs are canceled.

In the second scenario, when the training budget is not constrained, ABS-based pruning and dynamic mask training are conducted in order. In particular, we use ABS-based pruning for dynamic mask training as the criterion performs well provided long training duration (Huang et al., 2024). As for dynamic mask training, two specific cases involving full-parameter and LoRA training are considered. For full-parameter training, the SR-STE framework formulated using Eq. 3 is employed with one

Table 1: 64:2:5-sparse DeiT-small accuracy with update frequencies

| Update frequency | Top-1 Accu.(%) |
| --- | --- |
| Fixed | 78.72 |
| 1 | 72.96 |
| 5 | **79.65** |
| 10 | 79.48 |

modification. That is, the sparse masks are updated less frequently, specifically every five epochs instead of the one iteration per update reported in the original approach. As suggested in Table 1, this reduced update frequency enhances training stability, resulting in improved final accuracy for V:N:M sparse Transformers. Besides, we propose a three-staged LoRA training technique to train V:N:M-sparse Transformers under memory constraints, such as during the fine-tuning of LLMs. Overall, our scheme encompasses three training settings, each corresponding to one branch shown in Figure 2(a). For clarity, these three **t**raining **s**ettings are designated as **TS1**, **TS2**, and **TS3**, respectively. By

addressing all the possible conditions, our scheme significantly expands the applicability of V:N:M-sparse Transformers. Afterward, three key techniques marked with the blue color in Figure 2 in the scheme are detailed.

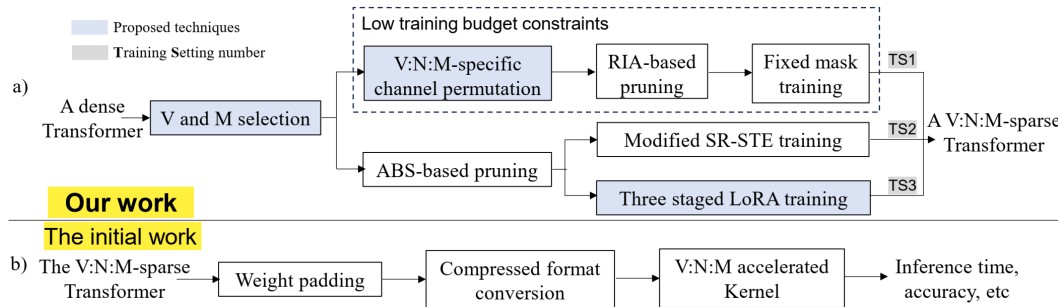

Figure 2: The workflow of utilizing V:N:M sparsity for Transformer inference acceleration. a) The generation process of a V:N:M sparse Transformer, which is our major contribution. b) The deployment process of the generated V:N:M-sparse Transformer.

## 4.1 V AND M SELECTION

For a dense Transformer, different V and M value combinations result in different final accuracy and speedups for its V:N:M-sparse counterparts. Among these combinations, we aim to select the proper V and M combinations with which a V:N:M-sparse Transformer consistently lies in the Pareto front in terms of both accuracy and speedups. However, it is often time-consuming to generate a V:N:M-sparse Transformer via sparse training, before its accuracy can be evaluated. To address this issue, we propose a heuristic V and M selection strategy including two key factors:

**1) Definition.** We define the process of solving for optimal combinations of $V$ and $M$ on the Pareto front as Eq. 4:

$$\arg\max_{V,M} Accu.\{f(\mathbf{w}(V, N, M)), \mathbf{d}_v\},$$

$$\text{subject to } Speedup\{f(\mathbf{w}), f(\mathbf{w}(V, N, M))\} \geq s \tag{4}$$

That is, given a specified speedup threshold $s$, training data $\mathbf{d}_t$, and a dense Transformer $f(\mathbf{w})$, our goal is to identify a proper V and M to maximize the accuracy of the Transformer's sparse version $f(\mathbf{w}(V, N, M))$, on validation data $\mathbf{d}_v$. This optimization is subject to the constraint that the speedup of $f(\mathbf{w}(V, N, M))$ relative to $f(\mathbf{w})$ is at least $s$. In practice, considering the GPU acceleration affinity, $\{V \in 2^k | k \in N^+, k \geq 4\}$ and $\{M \in N^+, M \geq 5\}$. Besides, $N \equiv 2$ in V:N:M sparsity if practical speedup is required.

Notably, the Pareto front is defined in our work as the optimization of accuracy subject to speedup constraints, rather than maximizing speedup under accuracy constraints. This distinction arises from our observation that the speedups of a V:N:M-sparse Transformer can be measured more rapidly than its accuracy. Thus, given a specified speedup threshold, it is feasible to quickly obtain all (V, M) combinations that lead to greater speedups.

**2) Sifting.** A two-phase sifting is conducted to select the optimal (V, M) combination from all the (V, M) combinations that meet the given speedup constraints. First, for a group of (V, M) combinations with the same V, it is evident the smallest M results in the highest accuracy of V:N:M-sparse Transformers, as a smaller M implies lower sparsity in the resulting sparse Transformers. This rule can be utilized to exclude most (V, M) combinations. Secondly, mask diversity (MD) (Hubara et al., 2021) is utilized to distinguish the rest of the (V, M) combinations. MD of V:N:M sparsity quantifies the number of unique masks permissible under the V:N:M sparse pattern constraint. Generally, a higher MD indicates greater sparse weight configuration flexibility, leading to better Transformer accuracy. Specifically, the MD of a V:N:M-sparse Transformer is:

$$MD_f = \prod_l MD_{V:N:M}^l, \qquad MD_{V:N:M}^l = [C_M^4 (C_4^N)^V]^{\frac{m}{V}\frac{n}{M}} = K(V, M)^{mn} \tag{5}$$

It is straightforward to prove that for the same Transformer, the relative order of different $MD_f$ is entirely determined by the values of V and M, i.e., $K$ in Eq. 5, and is irrespective of the weight shapes of the linear layers (See Appendix B for the proof). Thus, only calculating $K$ suffices for comparison and choosing the best (V, M) combination.

## 4.2 V:N:M-SPECIFIC CHANNEL PERMUTATION FOR LOW TRAINING BUDGET

To enhance the accuracy of V:N:M-sparse Transformer in scenarios with limited training budgets, i.e., only a small number of training epochs, a V:N:M-specific channel permutation (CP) approach is proposed and should be conducted before RIA-based pruning, i.e., as shown in Figure 2(a). Notably, CP for 2:4 and V:N:M sparsity is different. In 2:4 sparsity, only the input CP of a weight matrix influences the norm of importance scores of retained weights after pruning. In contrast, V:N:M sparsity allows both input and output CP to affect the retained norm. Specifically, both input and output CP for a weight matrix $\mathbf{W}$ are:

Table 2: 64:2:5-sparse DeiT-base accuracy on downstream tasks with different iterations

| Iterations | AVG Accu. (%) |
|---|---|
| 1 | 94.56 |
| 2 | **94.71** |
| 3 | 94.53 |
| 4 | 94.44 |

$$\mathbf{Y} = \mathbf{W}\mathbf{X} = \mathbf{P}_o^T \mathbf{P}_o \mathbf{W} \mathbf{P}_i \mathbf{P}_i^T \mathbf{X} = \mathbf{P}_o^T \mathbf{W}_p \mathbf{P}_i^T \mathbf{X}, \tag{6}$$

where $\mathbf{P}_o$ and $\mathbf{P}_i$ are output CP and input CP matrices, respectively. $\mathbf{W}_p$ is the weight matrix after CP. After conducting V:N:M-sparse pruning to the permuted $\mathbf{W}_p$, we aim for the norm of the importance scores of the retained weights to be maximized, thus the optimization objective is:

$$\arg\max_{\mathbf{P}_o, \mathbf{P}_i} \sum_{i,j} RIA_{ij}(S_{V:N:M}(\mathbf{W}_p)) \tag{7}$$

We employ alternative optimization to iteratively solve for $\mathbf{P}_o$ and $\mathbf{P}_i$. Specifically, both $\mathbf{P}_o$ and $\mathbf{P}_i$ are initialized as identity matrices. In the $k$th iteration,

$$\mathbf{P}_i^{k+1} = \arg\max_{\mathbf{P}_i} \sum_{i,j} RIA_{ij}(S_{V:N:M}(\mathbf{P}_o^k \mathbf{W} \mathbf{P}_i)) \tag{8}$$

$$\mathbf{P}_o^{k+1} = \arg\max_{\mathbf{P}_o} \sum_{i,j} RIA_{ij}(S_{V:N:M}(\mathbf{P}_o \mathbf{W} \mathbf{P}_i^k)) \tag{9}$$

Like (Zhang et al., 2024), Eq. 8 or 9 can be approximately modeled as the traditional linear sum assignment problem, efficiently solvable using Hungarian algorithm(Kuhn, 1955). The total number of iterations is 2, based on the ablation study presented in Table 2. Note that during inference, $\mathbf{P}_o^T$ and $\mathbf{P}_i^T$ can be fused with post-Layernorm or preceding linear layers in standard Transformers, generally resulting in negligible time overheads (Zhang et al., 2024).

## 4.3 THREE-STAGED LoRA TRAINING

For LoRA training (Hu et al., 2021), the V:N:M sparse version $\mathbf{W}'$ of a dense weight matrix $\mathbf{W}$ is derived by:

$$\mathbf{W}' = (\mathbf{W} + \mathbf{B}\mathbf{A}) \odot \mathbf{M}, \tag{10}$$

==where $\mathbf{B}$ and $\mathbf{A}$ are two low rank matrices. Normally, $\mathbf{M}$ is a function of $\mathbf{W}$, $\mathbf{B}$, and $\mathbf{A}$. During training, $\mathbf{W}$ remains fixed while $\mathbf{B}$ and $\mathbf{A}$ are updated.== We propose a three-stage LoRA training technique to enable dynamic mask training with LoRA and enhance the accuracy of V:N:M-sparse Transformers. 1) Dense LoRA. At the beginning of training, standard LoRA fine-tuning is applied to the Transformer, where the masks $\mathbf{M}$ are consistently all-one matrices. 2) Sparse LoRA with the dynamic masks. After the dense LoRA, the masks $\mathbf{M}$ are updated at regular intervals according to the V:N:M sparse patterns as training progresses, which means

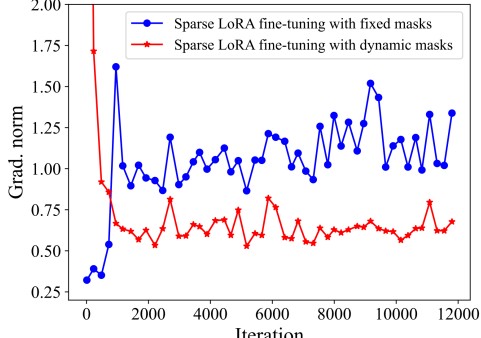

Figure 3: Gradient norm of Llama2-7B during sparse LoRA fine-tuning with dynamic masks and fixed masks, respectively.

each update occurs after a fixed number of iterations. During the mask update, the low-rank matrices **B** and **A** are merged with the original dense weight matrix **W** before calculating the importance scores and conducting V:N:M-sparse pruning. 3) Sparse LoRA with the fixed masks. While dynamic mask updates facilitate the exploration of appropriate V:N:M-sparse masks, they can also introduce instability in the training process. Furthermore, directly applying regularization such as SR-STE, as illustrated in Eq. 3, is challenging because the LoRA term **BA** is complex to regularize. Therefore, in the third stage, we advocate for fine-tuning with fixed masks to balance the exploration and exploitation of masks. At this stage, the masks **M** are inherited from the last update in the previous stage and remain unchanged until the training is completed.

It is important to note that the number of iterations in the first two stages should constitute a smaller proportion of the total iterations, with the majority allocated to the third stage, i.e., Sparse LoRA with fixed masks. This is because, in the absence of regularization, frequent updates to the masks can negatively impact the gradient flow of V:N:M sparse Transformers during fine-tuning. As exemplified in Figure 3, the gradient norms of LoRA with dynmaic masks are consistently lower than those with fixed masks. In practice, the iterations for the first two stages should not exceed 10% of the total iterations. More details about our technique are shown in Appendix C .

## 5 EXPERIMENTS

**Models, datasets and tasks** To evaluate the proposed V:N:M-sparse Transformer generation method—which incorporates V and M selection, V:N:M-specific channel permutations, and three staged LoRA training techniques, three benchmarks have been established: 1) DeiT (Touvron et al., 2021) for image classification. This benchmark is widely recognized for assessing the efficacy of model compression techniques in vision Transformers. Given that V:N:M sparsity operates at high sparsity levels (greater than 50%), the DeiT-tiny is excluded due to insufficient redundancy. The datasets used for the tasks include ImageNet-1K (Deng et al., 2009), Cifar-10 and Cifar-100 Krizhevsky et al. (2009), Bird and Vehicle from the subset of ImageNet-1K. Note that the latter four datasets are used to form downstream tasks. 2) Swin Transformers. This category of vision Transformers, known for its hierarchical architecture and shifted window mechanisms, demonstrates increased sensitivity to model compression (Liu et al., 2021). In this work, the V:N:M-sparse Swin Transformers are assessed across two tasks: image classification on the ImageNet-1K dataset and object detection on the COCO 2017 dataset (Lin et al., 2014). 3) Llama2-7B on downstream tasks including predicting the next token on wikitext2 (Merity et al., 2016) and eight well-established 5-shot tasks (Gao et al., 2021). In addition, the speedups of these V:N:M-sparse Transformers compared to their dense counterparts are measured on RTX 3090 GPUs, which were also the speed-testing platform in the initial study.

### 5.1 RESULTS FOR V AND M SELECTION

Figure 4 compares the accuracy and speedup for V:N:M-sparse Transformers, both with and without the proposed V and M selection technique. In this experiment, for practical speedups, V is confined to $[16, 32, 64, 128]$. Sparse Transformers with varying M values and unified V=64 are selected to establish the speedup threshold $s$, as specified in Eq. 4, with 64 representing a central value in the V distribution. Using the thresholds, the V and M selection technique is applied to derive new (V, M) and further generate the V:N:M-sparse Transformers accordingly, as indicated by the yellow lines in Figure 4. Specifically, in Figure 4(a), the V:N:M-sparse DeiT-base models located on the Pareto front exhibit higher Top-1 accuracy compared to the baseline, achieving a maximum accuracy improvement of 2.6% under similar speedup conditions. For the Llama2-7B, shown in Figure 4(b), the maximal average score difference is 3.41. Besides, when appropriate values of V and M are utilized, the sparse DeiT-base maintains a nearly lossless accuracy of 81.59% while achieving a speedup of 2.02 relative to its dense counterpart. Similarly, the sparse Llama2-7B achieves a speedup of 1.65, with a score of 50.85 on 5-shot tasks. Furthermore, it is essential to emphasize that V:N:M-sparse vision Transformers exhibit substantially greater speedups compared to 2:4 counterparts. As depicted in Figure 4(a), the 128:2:6-sparse DeiT-base matches the accuracy of the 2:4-sparse DeiT-base while achieving a **1.71x** speedup over its dense counterpart, in contrast to the 2:4-sparse version, which achieves only a 1.15x speedup. More speedup results of V:N:M-sparse Transformers are shown in Appendix D and E.

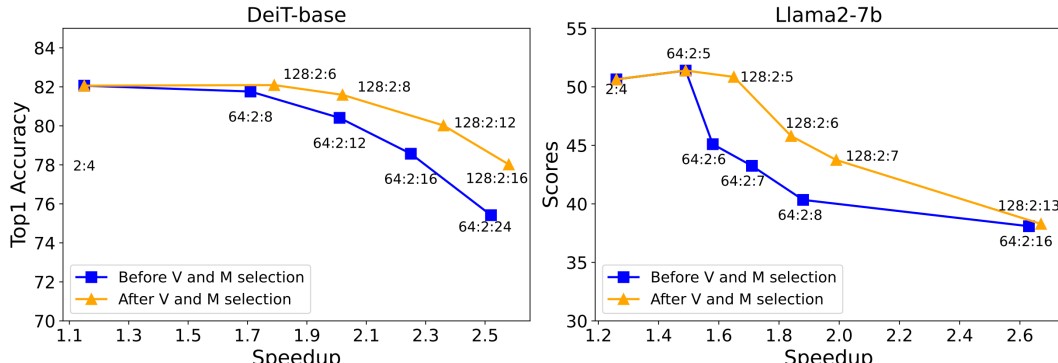

Figure 4: The speedup-accuracy curves of V:N:M-sparse Transformer. a) Top-1 accuracy of DeiT-base using TS2 with different V and M values. Accuracy of dense DeiT-base: 81.84%. b) Average scores of Llama2-7B using TS3 on 5-shot tasks with different V and M values. Average score of dense Llama2-7B: 61.99. Average score of 2:4-sparse Llama2-7B: 50.76. The speedup-accuracy curves using TS1 are shown in Figure 10 in Appendix F. More results for larger Transformers are shown in Figure 11, 12 and 13 in Appendix G, respectively.

## 5.2 RESULTS FOR V:N:M-SPECIFIC CHANNEL PERMUTATION AND TS1

Since the proposed V:N:M-specific CP is effective for low-budget training scenarios, we assess this technique on downstream tasks that allow for a limited number of training epochs to attain acceptable accuracy. The results, presented in Tables 3 and 4, were all obtained using TS1. For vision downstream tasks with a maximum of 30 training epochs, our technique enhances final accuracy by 0.71% for full-parameter training and 1.43% for LoRA training.

Table 3: 64:2:5-Sparse Llama2-7B results under 1500 training iterations.

| Pruning method | Wikitext2 PPL | 5-shot AVG scores |
|---|---|---|
| No | 11.19 | 48.44 |
| Ours | **11.09** | **49.69** |

Notably, when V:N:M-sparse pruning is only applied without any training, the accuracy gap can increase to 5.87% with or without our technique. For Llama2-7B, which underwent only 1500 training iterations—a relatively short duration compared to the 50,000 iterations shown in the subsequent Table 8, our technique improves the 5-shot scores by 1.25. More results are shown in Appendix I.

Table 4: Results of 64:2:5 DeiT-base on downstream tasks with 30 training epochs

| Downstream tasks | BIRD | | | VEHICLE | | | CIFAR-10 | | | CIFAR-100 | | | Average |
|---|---|---|---|---|---|---|---|---|---|---|---|---|---|
| Dense model (%) | 97.8 | | | 97.3 | | | 98.1 | | | 87.54 | | | Δ |
| Permutation Method | No | Ours | Δ | No | Ours | Δ | No | Ours | Δ | No | Ours | Δ | |
| Upon Pruning (%) | 81.6 | 87.6 | 6 | 78.6 | 85.7 | 7.1 | 84.42 | 89.85 | 5.43 | 57.65 | 63.07 | 5.42 | **5.87** |
| 30 epochs-all params. (%) | 96.5 | 96.9 | 0.4 | 95.7 | 96.4 | 0.7 | 97.51 | 97.73 | 0.22 | 85.54 | 86.65 | 1.11 | **0.71** |
| 30 epochs-LoRA (%) | 96.3 | 96.9 | 0.6 | 94.7 | 96.4 | 1.7 | 97.26 | 97.73 | 0.47 | 83.96 | 86.65 | 2.69 | **1.63** |

## 5.3 RESULTS FOR TS2

**DeiT on image classification** As the training budget is not limited in terms of TS2, the experiments are to investigate the extent to which the DeiT can be compressed while still achieving nearly lossless performance, i.e., gap$< 0.3\%$. As shown in Table 5, our TS2 allows DeiT-base to achieve lossless accuracy at a sparsity level of 75%, represented as 64:2:8. This level of sparsity results in a 73.8% reduction in parameters and a 71.6% reduction in FLOPs. Similarly, DeiT-small maintains lossless accuracy at 64:2:5, achieving a 57.9% reduction in parameters and a 54.3% reduction in FLOPs. Due to computational power limitations, larger Transformers, such as ViT-huge and ViT-giant, were not included in this investigation. However, it is generally acknowledged that larger models tend to exhibit greater redundancy. For the ImageNet-1K classification task, larger vision Transformers than DeiT-base are anticipated to achieve higher sparsity than 64:2:8 while maintaining lossless accuracy.

Table 5: Results comparison of sparse DeiTs on ImageNet-1K. See Appendix J for detailed description of (↓%), Δ, *, and related works.

| Model | Param. (M) | (↓%) | FLOPs (G) | (↓%) | Top-1 Acc. (%) | Δ |
|---|---|---|---|---|---|---|
| DeiT-B | 86.6 | - | 17.6 | - | 81.84 | - |
| DeiT-B-600e | 86.6 | - | 17.6 | - | 82.01 | +0.17 |
| T2T-ViT-24 | 64.1 | 26.0 | 13.8 | 21.6 | 82.30 | +0.46 |
| PVT-L | 61.4 | 29.1 | 9.8 | 44.3 | 81.70 | -0.14 |
| AutoFormer-B | 54.0 | 37.6 | 11.0 | 37.5 | 82.40 | +0.56 |
| SSP-B | 56.8 | 34.4 | 11.8 | 33.1 | 80.80 | -1.04 |
| $S^2$ViTE-B | 56.8 | 34.4 | 11.8 | 33.1 | 82.22 | +0.38 |
| Evo-ViT | 56.3 | - | 11.7 | 33.3 | 80.30 | -0.54 |
| EViT-DeiT-B | 86.6 | - | 11.5 | 34.7 | 81.50 | -0.34 |
| DynamicViT | 61.0 | - | 11.5 | 34.7 | 81.30 | -0.54 |
| ViT-Slim | 52.6 | 39.3 | 10.6 | 39.6 | 82.40 | +0.56 |
| VTP-B | 47.3 | 45.4 | 10.0 | 43.2 | 80.70 | -1.14 |
| PS-ViT-B | 86.6 | - | 9.8 | 44.3 | 81.50 | -0.34 |
| UVC | - | - | 8.0 | 54.5 | 80.57 | -1.27 |
| SAViT | 25.4 | 70.7 | 5.3 | 69.9 | 81.66 | -0.18 |
| NViT-B (ASP) | 17 | 80.4 | 6.8 | 61.4 | 83.29 | -0.07* |
| Ours (64:2:8) | 22.7 | 73.8 | 5.0 | 71.6 | 81.08 | -0.76 |
| Ours-600e (64:2:8) | **22.7** | **73.8** | **5.0** | **71.6** | **81.76** | **-0.08** |
| DeiT-S | 22.1 | - | 4.6 | - | 79.85 | - |
| DeiT-S-600e | 22.1 | - | 4.6 | - | 80.02 | +0.17 |
| AdaViT-S | 22.1 | 0.0 | 3.6 | 21.7 | 78.60 | -1.25 |
| DynamicViT | 22.1 | - | 3.4 | 26.1 | 79.60 | -0.25 |
| EViT-DeiT-S | 22.1 | - | 3.4 | 34.8 | 79.50 | -0.35 |
| SSP-S | 14.6 | 33.3 | 3.1 | 31.6 | 77.74 | -2.11 |
| $S^2$ViTE-S | 14.6 | 33.3 | 3.1 | 31.6 | 79.22 | -0.63 |
| SAViT | 14.7 | 33.5 | 3.1 | 31.7 | 80.11 | +0.26 |
| NViT-S (ASP) | 10.5 | 52.5 | 4.2 | 8.7 | 82.19 | +0.99* |
| Ours (64:2:5) | 9.3 | 57.9 | 2.1 | 54.3 | 78.97 | -0.68 |
| Ours-600e (64:2:5) | **9.3** | **57.9** | **2.1** | **54.3** | **79.65** | **-0.2** |

**Swin Transformers** The Swin Transformer is generally considered challenging to compress. However, the V:N:M-sparse Swin Transformer, utilizing our TS2, achieves results comparable to the state of the art. As shown in Table 6, under identical training epochs, the Swin Transformer at a sparsity of 32:2:5 achieves the same Top-1 accuracy as LPViT (Xu et al., 2024). Notably, it achieves a 59.1% reduction in parameters and a 60.4% reduction in FLOPs, which is significantly greater than LPViT's 27% reduction in FLOPs relative to its dense counterpart. For object detection, the dense H-DETR, using Swin-Tiny as the backbone and trained for 24 epochs, is employed to generate the V:N:M-sparse H-DETR. With TS2 and 12 training epochs, the H-DETR at 32:2:5 achieves a mean Average Precision (mAP) of 47.8%, outperforming the dense equivalent trained for 12 epochs, by 2.5%. Furthermore, at a sparsity of 32:2:6, the sparse H-DETR with TS2 retains an mAP of 45.3%, surpassing that with the fixed mask training (FMT) setting by 1.2%. These results demonstrate that our TS2 is more favorable to the accuracy restoration of V:N:M-sparse Transformers.

Table 6: V:N:M-sparse Swin Transformers on image classification and object detection. See Appendix J for detailed description of (↓%), Δ and related works.

| Model | Method | Param.(M) | (↓%) | FLOPs (G) | (↓%) | Top-1 Acuu. (%) |
|---|---|---|---|---|---|---|
| Swin-base | Dense | 87.8 | - | 15.4 | - | 83.51 |
| | LPViT | 64.1* | 27.0 | 11.24 | 27.0 | 81.7 |
| | Ours(32:2:5) | **35.9** | **59.1** | **6.1** | **60.4** | **81.7** |
| H-DETR (Swin-Tiny) | Dense | 40.2 | - | 212.0 | - | 45.3 |
| | Dense-24e | 40.2 | - | 212.0 | - | 49.2 |
| | FMT(32:2:5) | 18.1 | 55.0 | 93.7 | 55.8 | 47.5 |
| | Ours(32:2:5) | **18.1** | **55.0** | **93.7** | **55.8** | **47.8** |
| | FMT(32:2:6) | 15.5 | 61.4 | 78.0 | 63.2 | 44.1 |
| | Ours(32:2:6) | **15.5** | **61.4** | **78.0** | **63.2** | **45.3** |

## 5.4 Results for three-staged LoRA training and TS3

The experiments aim to demonstrate that the V:N:M-sparse Llama2, with our TS3 involving three-staged LoRA training, can achieve performance levels comparable to its 2:4-sparse version formed by post-pruning approaches, e.g., RIA (Zhang et al., 2024). The LoRA training was conducted over 50,000 samples, each consisting of 1,024 tokens. Each training iteration utilizes one sample as input, resulting in a total of 50,000 iterations. The results show that utilizing our approach, Llama2-7B achieved a perplexity (PPL) of 9.97 on the Wikitext2 dataset, as shown in Table 7. Additionally, in the 5-shot tasks, the 64:2:5-sparse Llama2-7B scored 53.04, outperforming the state-of-the-art RIA-based post-pruning 2:4-sparse counterpart, which yielded a score of 50.64, as shown in Table 8. Furthermore, the 64:2:5-sparse Llama2-7B delivers a higher speedup compared to the 2:4 sparsity, achieving a speedup of **1.49** versus 1.26.

Table 7: 64:2:5-Sparse Llama2-7B results on Wikitext2. Training iteration: 50000

|  | PPL | Speedup |
| --- | --- | --- |
| Dense | 5.12 | 1 |
| SparseGPT (2:4) | 10.17 | 1.26 |
| Wanda (2:4) | 11.27 | 1.26 |
| RIA (2:4) | 10.52 | 1.26 |
| Ours (64:2:5) | **9.97** | **1.49** |

Table 8: 5-shot results of 64:2:5-sparse Llama2-7B

|  | OpenBookQA | ARC-C | ARE-E | WinoGrande | Hellaswag | RTE | PIQA | BoolQ | AVG |
| --- | --- | --- | --- | --- | --- | --- | --- | --- | --- |
| Dense | 31.40 | 43.43 | 76.26 | 69.06 | 57.23 | 62.82 | 78.08 | 77.74 | 61.99 |
| Wanda (2:4) | 20.00 | 30.97 | 65.24 | 59.75 | 39.88 | 54.51 | 69.53 | 66.21 | 50.76 |
| RIA (2:4) | 20.20 | 30.80 | 64.77 | 59.59 | 40.63 | 55.23 | 69.70 | 64.19 | 50.64 |
| Ours (64:2:5) | 25.80 | 34.30 | 64.27 | 61.80 | 42.88 | 56.68 | 72.03 | 66.54 | **53.04** |

**Ablation study** For TS3, five different LoRA training strategies are explored: A) Sparse LoRA with fixed masks; B) Dense LoRA combined with sparse LoRA, using fixed masks; C) Sparse LoRA with dynamic masks, updated at equal intervals; D) Sparse LoRA with dynamic masks, utilizing early updates only; E) Our proposed three-stage training technique. Figure 5 illustrates that the proposed technique consistently achieves the highest scores across various sparsity levels, thanks to its optimal balance between enhanced gradient flow and moderate mask exploration. In contrast, Scheme C, which employs dynamic mask updates at equal intervals throughout the training, produces the lowest scores due to impaired gradient flow and increased training instability. Additionally, only using one or two stages within our technique for LoRA training results in suboptimal performance. It is evident that the three-staged LoRA training is the most effective technique for V:N:M-sparse Transformers.

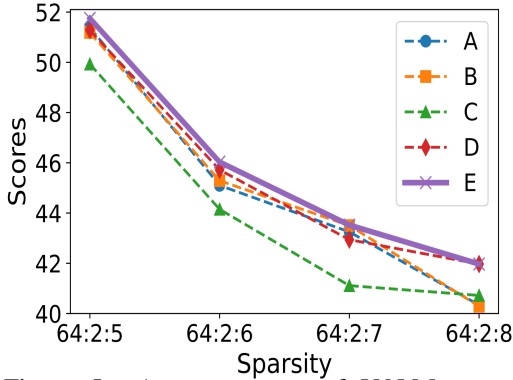

Figure 5: Average scores of V:N:M-sparse Llama2-7B on 5-shot tasks under different LoRA finetuning schemes. Dense: 61.99. 2:4-sparse: 50.76

## 6 Conclusion

This study focuses on enhancing the accuracy and accuracy-speedup trade-offs of V:N:M-sparse Transformers in multiple scenarios. We address the crucial yet unexplored questions specific to V:N:M sparsity, including selecting appropriate values for V and M, and CP tailored for V:N:M sparsity. Additionally, we propose a three-staged LoRA training technique, which for the first extends V:N:M sparsity to LLMs. Extensive experiments demonstrate that, with our methodology, V:N:M-sparse Transformers can attain nearly lossless accuracy or perform comparably to those with post-pruning 2:4 sparsity. Given its superior speed performance, we conclude that V:N:M sparsity is more effective than 2:4 for compressing highly redundant Transformers in inference-cost-sensitive scenarios. We hope our work to promote the widespread use of V:N:M sparsity as a truly effective solution for compressing Transformers.

**Reproducibility Statement** We are committed to ensuring the reproducibility of our work. The theoretical foundations and assumptions underlying our framework are thoroughly discussed in Section 4, and some proofs of our claims are provided in the appendix. Detailed descriptions of our experiments, including the architecture configurations and hyperparameters used for training the V:N:M-sparse Transformers, can be found in Section 5 of the main text. In addition, We will publicly release our source code anonymously at the appropriate time. We believe these resources collectively facilitate the reproducibility of our findings and ensure that our methodologies can be adopted in future research without difficulty.

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

# APPENDIX

## A  V:N:M SPARSITY DETAILS

As illustrated in Figure 6(a), the sparse weight matrix $\mathbf{A}$ is transformed into three smaller, more compact matrices: $\mathbf{A}_n$, $\mathbf{A}_{i1}$, and $\mathbf{A}_{i2}$. The matrix $\mathbf{A}_n$ contains the non-zero values from the sparse matrix $\mathbf{A}$, preserving their relative positions; that is, non-zero values in the same rows or columns of $\mathbf{A}$ remain in the same rows or columns in $\mathbf{A}_n$. The matrix $\mathbf{A}_{i1}$ lists the indices of the four columns within each block of $\mathbf{A}_n$ that are designated for 2:4 sparsity. Meanwhile, $\mathbf{A}_{i2}$ mirrors the shape of $\mathbf{A}_n$, with each non-zero value in $\mathbf{A}_n$ having a corresponding index in $\mathbf{A}_{i2}$ that indicates its position among four consecutive elements. Figure 6(b) further illustrates the process of V:N:M-sparse matrix multiplication (MM) using sparse tensor cores. The primary function of the V:N:M-sparse MM kernel is to retrieve the retained weights and the corresponding tiles of the input matrix $\mathbf{B}$. This is done to align the data layout with that of a 2:4-sparse MM. By doing so, the sparse tensor core can efficiently compute the output $\mathbf{C}$ in chunks.

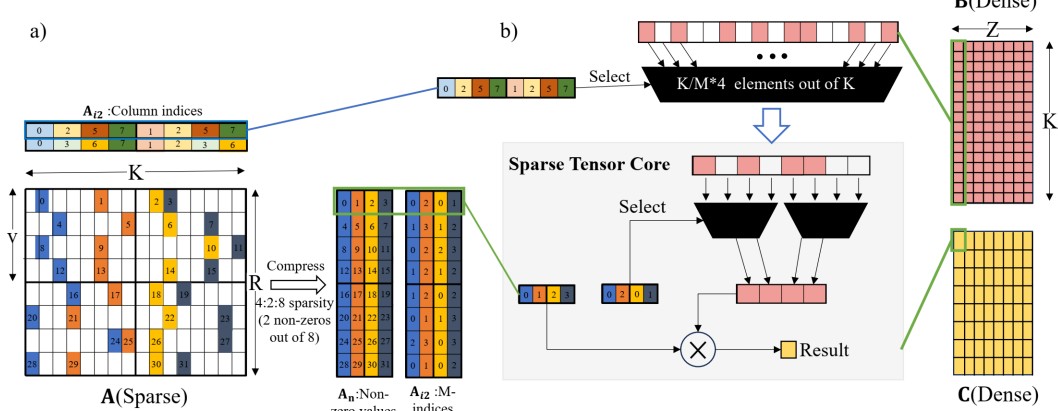

Figure 6: Details of V:N:M sparsity. a) An example of converting a V:N:M-sparse matrix to a compressed format. b) A schematic illustrating the hardware operations for V:N:M-sparse matrix multiplications.

## B  PROOF OF THE RELATIVE ORDER FOR MASK DIVERSITY

Suppose we have two different V:N:M sparsity patterns for the same Transformer, adopting different values of $V_1$, $M_1$ and $V_2$, $M_2$, while N remains constant at $N \equiv 2$. In this case, for a linear layer, where the shape of the linear weight is $[m, n]$,

$$\frac{MD^l_{V_1:N:M_1}}{MD^l_{V_2:N:M_2}} = \frac{[C^4_{M_1}(C^2_4)^{V_1}]^{\frac{m}{V_1} \cdot \frac{n}{M_1}}}{[C^4_{M_2}(C^2_4)^{V_2}]^{\frac{m}{V_2} \cdot \frac{n}{M_2}}}$$

$$= \left\{ \frac{[C^4_{M_1}(C^2_4)^{V_1}]^{\frac{1}{V_1 M_1}}}{[C^4_{M_2}(C^2_4)^{V_2}]^{\frac{1}{V_2 M_2}}} \right\}^{mn}$$

$$\equiv \frac{K^{mn}_1}{K^{mn}_2}$$

Thus, for a complete network, suppose the linear layer $l$ has a linear weight shape of $[m_l, n_l]$. We calculate the ratio of the MD under two different selections of the V and M parameters for the same model. The ratio $\frac{MD_1}{MD_2}$ can be expressed as follows:

$$\frac{MD_1}{MD_2} = \frac{\prod_l MD^l_{V_1:N:M_1}}{\prod_l MD^l_{V_2:N:M_2}}$$

$$= \frac{\prod_l K^{m_l n_l}_1}{\prod_l K^{m_l n_l}_2}$$

$$= \left(\frac{K_1}{K_2}\right)^{\sum_l m_l n_l}$$

This indicates that for the same Transformer, as long as $K_1 > K_2$, it implies that the overall network's masking diversity satisfies $MD_1 > MD_2$, regardless of the specific shapes of the linear weights.

Besides, here we would like to further clarify that MD, rather than model parameter counts, serves as a better indicator for V:N:M-sparse Transformers. We present a specific example in Table 9. Among three (V, M) configurations including (16, 16), (32, 16) and (128, 15), the 128:2:15-sparse DeiT-base has the highest parameter count, yet the 16:2:16-sparse DeiT-base, which exhibits the highest MD, achieves the best accuracy, significantly surpassing the other two configurations. This principle highlights that **both the sparse granularity determined by V and the retained parameter counts dictated by M influence the accuracy of V:N:M-sparse Transformers**. Compared to relying solely on parameter counts, MD accounts for both factors, thereby providing a more accurate measure of performance.

Table 9: V:N:M-sparse DeiT-Base's accuracy of 30-epochs LoRA training on downstream tasks.

| (V,M) | (16,16) | (32,16) | (128,15) |
|---|---|---|---|
| Params.(M) | 10.8 | 10.8 | **11.5** |
| Simplified MD | 20837 | 18674 | 18168 |
| Bird Accuracy (%) | 84.1 | 82.1 | 79.5 |
| Vehicle Accuracy (%) | 77.4 | 75.2 | 72.6 |
| CIFAR10 Accuracy (%) | 86.5 | 85.4 | 84.0 |
| CIFAR100 Accuracy (%) | 55.9 | 53.3 | 50.5 |
| Average Accuracy (%) | **76.0** | 74 | 71.7 |

## C  DETAILS OF OUR THREE STAGED LoRA TRAINING

We would like to detail the configurations used in our three-stage LoRA training as outlined in Table 10. 1) To obtain the initial dynamic sparse masks $M$ for the weight matrix $W$ in the second stage of

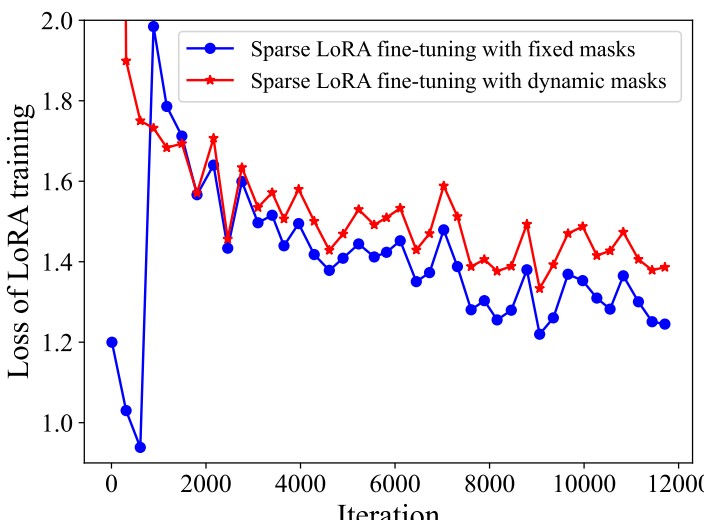

Figure 7: Loss of Llama2-7B during sparse LoRA fine-tuning with dynamic masks and fixed masks, respectively.

LoRA training, we first merge the LoRA matrices $BA$ with $W$. RIA-based pruning is then applied to the merged matrix, where the retained weights are accordingly assigned a value of 1 in $M$, while the pruned weights are assigned a value of 0 in $M$. 2) We set the interval for updating sparse masks to 20 iterations. A smaller interval can destabilize LoRA training, a phenomenon also noted in DeiT-base in our paper (Please refer to Table 1 in our paper). 3) We adjust hyperparameters, including the mask update intervals and update counts, to ensure that the first, second, and third stages account for approximately 2.5%, 2.5%, and 95% of the total training, respectively. 4) Following standard practice, we use 1,024 tokens for each training iteration, resulting in a total of 12 million tokens for our LoRA training. 5) The ranks of the LoRA matrices, specifically $A$ and $B$, are set to 16, with LoRA $\alpha$ configured to 32 to maintain the regular setting of $\alpha/\text{rank} = 2$. 6) All the linear layers in Llama2 are equipped with LoRA training.

Besides, to further intuitively illustrate the necessity for infrequent mask updates, we present the loss change curves for sparse LoRA fine-tuning with both fixed and dynamic masks. As depicted in Figure 7, constant mask updates result in a progressively higher training loss compared to the fixed-mask training as the training progresses.

Table 10: Details of our three-staged LoRA training

| Items | Settings |
| --- | --- |
| Initial of dynamic sparse masks | RIA-based pruning |
| Interval of updating sparse masks | 20 iterations per update |
| Actual training iteration assignment | (A total of 12000) First stage: 320; Second stage: 320; Third stage: 11360 |
| Overall LoRA training tokens | 1024 tokens $\times$ 12000 iterations = 12 million tokens |
| LoRA rank | 16 |
| LoRA $\alpha$ | 32 |
| LoRA modules in Llama2 | q_porj, k_proj, v_proj, o_proj, up_proj, gate_proj, down_proj |

# D    DETAILED SPEEDUP RESULTS OF V:N:M-SPARSE TRANSFORMERS

We present additional end-to-end speedup results for V:N:M-sparse Transformers, as illustrated in Figures 8 and 9. It is noteworthy that permutation overheads are excluded from these measurements, as the permutation operator can theoretically be fused with matrix multiplication or LayerNorm operations, as already demonstrated by (Zhang et al., 2024). For the Llama2 model, we find the speedup increases monotonically as the values of V and M grow larger. In contrast, the speedup for the DeiT series shows some fluctuations, primarily due to the additional inference time overheads caused by weight padding, which slightly reduces the overall speedup for smaller Transformers. However, as the depth and width of the Transformer increase, these fluctuations diminish rapidly.

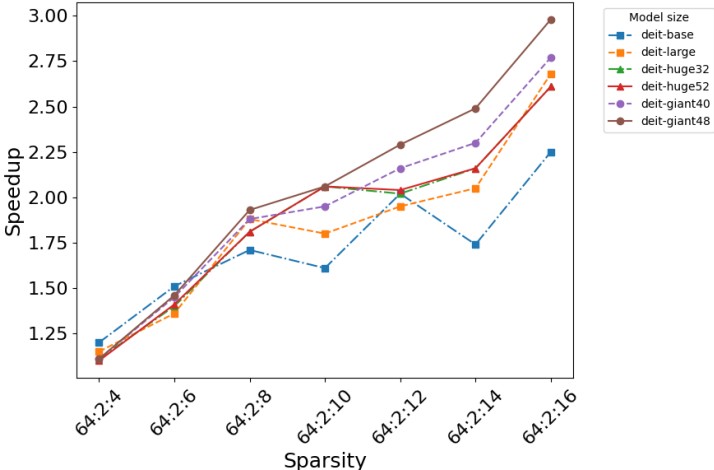

Figure 8: The Speedup of different model sizes (deit-large, huge, giant) under different sparsity. Here the speedup of dense models: 1.

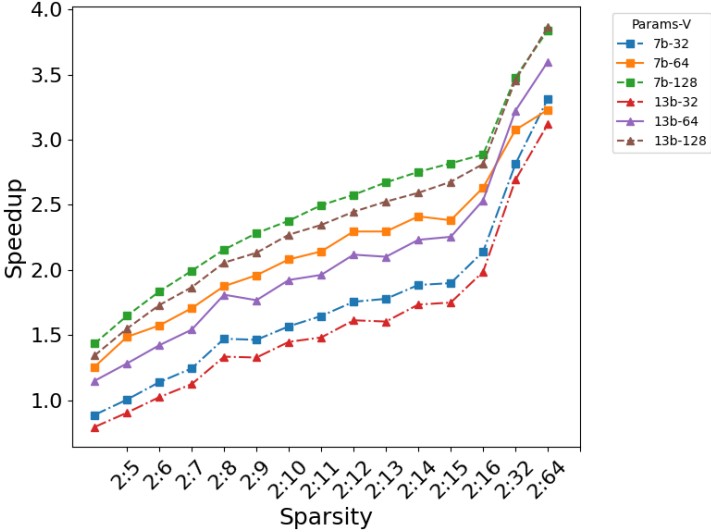

Figure 9: The Speedup of Llama2 models under different sparsity. Here the speedup of dense models: 1.

## E  V AND M SELECTION UNDER DIFFERENT BATCH SIZES

When varying the batch size, the selected values for V and M differ accordingly. For example, the results for Llama2-7B at various batch sizes are shown in Table 11. The speedup thresholds are randomly generated from the range [1, 2.5]. Since the speedup of a V:N:M-sparse Transformer compared to its dense counterpart varies with different batch sizes, the selection results will also differ. We consider this phenomenon to be normal and it does not impact our technical contributions. In practical LLM inference systems, especially in cloud environments, multiple queries are often concatenated into a fixed batch before inference, which scenario is still suitable for the application of our techniques.

Table 11: Selected V and M values of Llama2-7B under different batch sizes (BS). (X,4) means 2:4 sparsity.

| | Speedup threshold | 1.14 | 1.26 | 1.34 | 1.52 | 1.65 | 1.88 | 2.12 |
|---|---|---|---|---|---|---|---|---|
| BS=1 | Selected (V,M) | (X, 4) | (X, 4) | (64, 5) | (128, 5) | (128, 5) | (128, 7) | (128, 8) |
| | Speedup | 1.26 | 1.26 | 1.49 | 1.65 | 1.65 | 1.99 | 2.16 |
| BS=2 | Selected (V,M) | (X, 4) | (64, 5) | (64, 5) | (128, 5) | (128, 6) | (128, 8) | (128, 10) |
| | Speedup | 1.19 | 1.36 | 1.36 | 1.54 | 1.7 | 2.01 | 2.17 |
| BS=4 | Selected (V,M) | (64, 5) | (64, 5) | (128, 5) | (128, 6) | (128, 7) | (128, 8) | (128, 11) |
| | Speedup | 1.26 | 1.26 | 1.45 | 1.6 | 1.74 | 1.88 | 2.12 |
| BS=8 | Selected (V,M) | (64, 5) | (128, 5) | (128, 5) | (128, 6) | (128, 7) | (128, 9) | (128, 13) |
| | Speedup | 1.14 | 1.34 | 1.34 | 1.55 | 1.68 | 1.88 | 2.14 |
| BS=16 | Selected (V,M) | (64, 5) | (128, 5) | (128, 6) | (128, 7) | (128, 8) | (128, 11) | (128, 13) |
| | Speedup | 1.14 | 1.29 | 1.41 | 1.55 | 1.72 | 1.91 | 2.12 |

## F  SPEEDUP-ACCURACY CURVES OF DEIT-BASE ON DOWNSTREAM TASKS

Our V and M selection technique is applicable across all three training settings: TS1, TS2, and TS3. Therefore, it is essential to also evaluate our method using TS1, as shown in Figure 10. We present the average Top-1 accuracy of the DeiT-base model on the Bird, Vehicle, Cifar-10, and Cifar-100 datasets for different V and M values. The results clearly demonstrate that our V and M selection enhances the speedup-accuracy trade-off for the DeiT-base on downstream tasks. Notably, to achieve acceptable accuracy in these tasks, it is recommended that the V:N:M sparsity not be set too high.

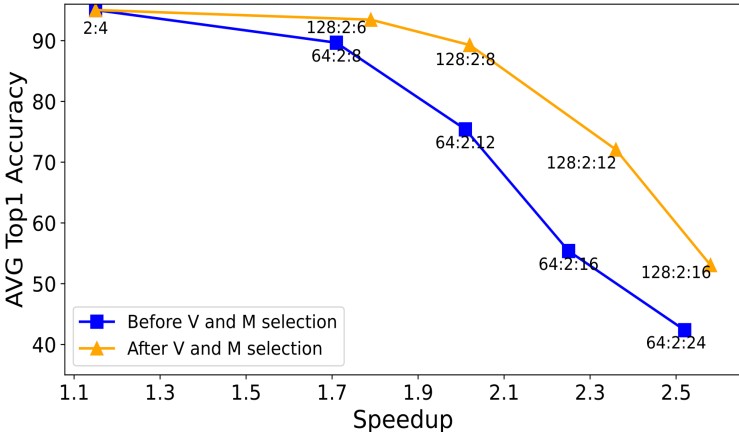

Figure 10: Average Top-1 accuracy of DeiT-base on downstream tasks with different V and M vlaues. 

## G SPEEDUP-ACCURACY CURVES OF LARGER V:N:M-TRANSFORMERS

To further validate our method on larger-scale Transformers, we have extended our proposed V and M selection method to three additional representative models: ViT-large, ViT-huge, and Llama2-13B. Notably, Llama2-13B undergoes LoRA training on the same datasets as Llama2-7B, specifically Wikitext2 and Alpaca (Taori et al., 2023), which aligns with standard practices in the fine-tuning of LLMs. The experimental results, presented in Figures 11, 12, and 13, respectively, demonstrate that V:N:M-sparse Transformers employing our selection method consistently achieve superior accuracy-speedup trade-offs compared to those that do not. It is clear that our V and M selection method significantly enhances these trade-offs, even for large-scale V:N:M-sparse Transformers.

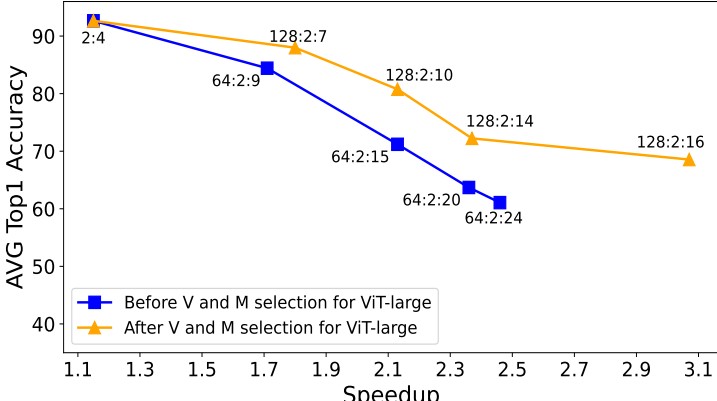

Figure 11: Average Top-1 accuracy of ViT-large on downstream tasks with different V and M vlaues. Average Top-1 accuracy of dense ViT-large: 94.5%.

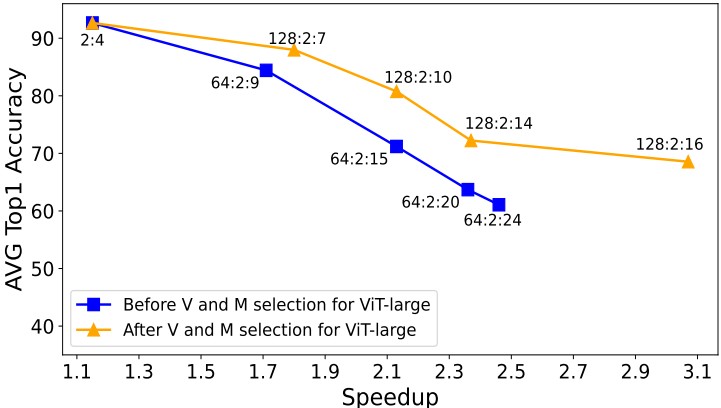

Figure 12: Average Top-1 accuracy of ViT-huge on downstream tasks with different V and M vlaues. Average Top-1 accuracy of dense ViT-huge: 93%.

## H THE ROLE OF RIA-BASED PRUNING IN OUR FRAMEWORK

Table 12 demonstrates that, under limited training budget constraints, RIA-based pruning is more effective than ABS-based pruning for improving the accuracy of V:N:M-sparse Transformers on downstream tasks. This advantage holds true regardless of whether the V:N:M-sparse Transformers are obtained through LoRA or full-parameter training. Specifically, RIA-based pruning combined with our CP technique can enhance the accuracy of a 64:2:5 DeiT-base Transformer by up to 1.63%

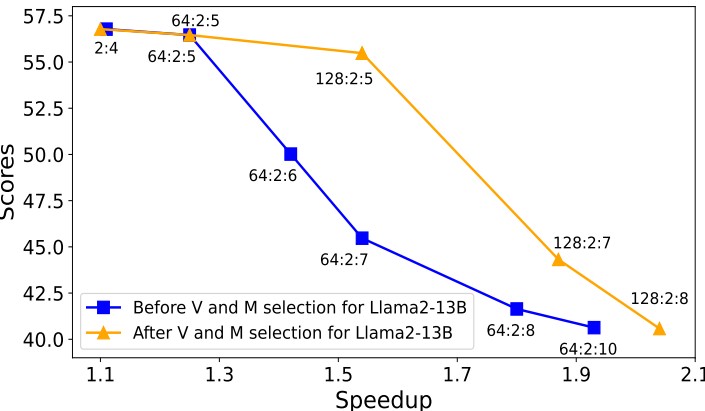

Figure 13: Average scores of Llama2-13B using TS3 on 5-shot tasks with different V and M values. Average score of dense Llama2-13B: 68.23. Average score of 2:4-sparse Llama2-7B: 56.78.

after 30 epochs of LoRA training. In contrast, ABS-based pruning only achieves a 0.52% accuracy improvement for the same model and training duration.

Table 12: Performance of 64:2:5-sparse DeiT-base on downstream tasks with RIA and ABS -based pruning, respectively.

| Downstream tasks | | BIRD | | | VEHICLE | | | CIFAR-10 | | | CIFAR-100 | | | Average |
|---|---|---|---|---|---|---|---|---|---|---|---|---|---|---|
| Dense model(%) | | 97.8 | | | 97.3 | | | 98.1 | | | 87.54 | | | |
| Permutation Method | Importance Score | No | Ours | Δ | No | Ours | Δ | No | Ours | Δ | No | Ours | Δ | |
| Upon Pruning (%) | | 81.6 | 87.6 | 6.0 | 78.6 | 85.7 | 7.1 | 84.42 | 89.85 | 5.43 | 57.65 | 63.07 | 5.42 | **5.87** |
| 30 epochs-all params (%) | RIA | 96.5 | 96.9 | 0.4 | 95.7 | 96.4 | 0.7 | 97.51 | 97.73 | 0.22 | 85.54 | 86.65 | 1.11 | **0.71** |
| 30 epochs-LoRA (%) | | 96.3 | 96.9 | 0.6 | 94.7 | 96.4 | 1.7 | 97.26 | 97.73 | 0.47 | 83.96 | 86.65 | 2.69 | **1.63** |
| Upon Pruning (%) | | 41.2 | 43.1 | 1.9 | 29.4 | 43.4 | 14.0 | 37.09 | 35.40 | -1.69 | 10.70 | 15.01 | 4.31 | **4.63** |
| 30 epochs-all params (%) | ABS | 96.0 | 95.9 | -0.1 | 95.4 | 95.3 | -0.1 | 97.39 | 97.50 | 0.11 | 84.83 | 85.63 | 0.80 | **0.18** |
| 30 epochs-LoRA (%) | | 95.9 | 95.8 | -0.1 | 94.1 | 95.4 | 1.3 | 96.68 | 97.03 | 0.35 | 82.80 | 83.35 | 0.55 | **0.52** |

## I EFFECTIVENESS OF V:N:M-SPECIFIC CP UNDER DIFFERENT SPARSITY

To further illustrate the effectiveness of our V:N:M-specific CP, we present additional results regarding CP performance under various V:N:M ratios in the limited-training-budget scenario, as shown in Tables 13, 14, and 15. The results clearly indicate that our CP method significantly enhances the accuracy of these V:N:M-sparse Transformers. Besides, our CP typically brings more accuracy gains for higher sparsity, e.g., achieving an improvement of 4.18% at 64:2:16 compared to an increase of 1.63% at 64:2:5 after 30 epochs of LoRA training.

Table 13: Results of 64:2:16 DeiT-base on downstream tasks with 30 training epochs.

| Downstream tasks | BIRD | | | VEHICLE | | | CIFAR-10 | | | CIFAR-100 | | | Average |
|---|---|---|---|---|---|---|---|---|---|---|---|---|---|
| Dense model (%) | 97.7 | | | 97.3 | | | 98.1 | | | 87.54 | | | |
| Permutation Method | No | Ours | Δ | No | Ours | Δ | No | Ours | Δ | No | Ours | Δ | |
| Upon Pruning (%) | 7.40 | 8.10 | 0.6 | 5.00 | 7.00 | 2.0 | 10.8 | 12.4 | 1.6 | 1.00 | 1.35 | 0.35 | **1.14** |
| 30 epochs-all params. (%) | 84.9 | 89.9 | 5.0 | 77.8 | 83.5 | 5.7 | 89.9 | 92.7 | 2.8 | 64.4 | 70.7 | 6.3 | **4.95** |
| 30 epochs-LoRA (%) | 79.7 | 84.1 | 4.3 | 74.3 | 77.4 | 3.1 | 82.9 | 86.5 | 3.6 | 50.2 | 55.9 | 5.7 | **4.18** |

## J DETAILS OF SIGNS AND RELATED WORKS IN TABLE 5 AND 6

Table 5 and 6 have the similar headers. In the headers, the first ↓ % always indicates the proportion of parameter reduction relative to the parameters of the dense model, while the second ↓ % signifies

Table 14: Results of 32:2:16 DeiT-base on downstream tasks with 30 training epochs.

| Downstream tasks | BIRD | | | VEHICLE | | | CIFAR-10 | | | CIFAR-100 | | | Average |
|---|---|---|---|---|---|---|---|---|---|---|---|---|---|
| Dense model (%) | 97.7 | | | 97.3 | | | 98.1 | | | 87.54 | | | |
| Permutation Method | No | Ours | Δ | No | Ours | Δ | No | Ours | Δ | No | Ours | Δ | |
| Upon Pruning (%) | 8.10 | 6.10 | -2.0 | 6.60 | 6.20 | -0.4 | 9.98 | 13.6 | 3.6 | 1.24 | 1.21 | -0.03 | **0.29** |
| 30 epochs-all params. (%) | 83.1 | 87.8 | 4.7 | 75.0 | 81.2 | 6.2 | 88.1 | 91.6 | 3.5 | 61.2 | 67.4 | 6.2 | **5.15** |
| 30 epochs-LoRA (%) | 78.8 | 82.1 | 3.3 | 72.9 | 75.2 | 2.3 | 81.2 | 85.4 | 4.2 | 48.2 | 53.3 | 5.1 | **3.73** |

Table 15: Results of 128:2:15 DeiT-base on downstream tasks with 30 training epochs.

| Downstream tasks | BIRD | | | VEHICLE | | | CIFAR-10 | | | CIFAR-100 | | | Average |
|---|---|---|---|---|---|---|---|---|---|---|---|---|---|
| Dense model (%) | 97.7 | | | 97.3 | | | 98.1 | | | 87.54 | | | |
| Permutation Method | No | Ours | Δ | No | Ours | Δ | No | Ours | Δ | No | Ours | Δ | |
| Upon Pruning (%) | 5.00 | 4.50 | -0.5 | 6.60 | 7.10 | 0.5 | 10.2 | 12.4 | 2.2 | 0.97 | 1.32 | 0.35 | **0.64** |
| 30 epochs-all params. (%) | 83.2 | 86.0 | 2.8 | 74.0 | 78.8 | 4.8 | 88.1 | 91.0 | 2.9 | 60.8 | 65.6 | 4.8 | **3.83** |
| 30 epochs-LoRA (%) | 77.2 | 79.5 | 2.3 | 70.4 | 72.6 | 2.2 | 79.7 | 84.0 | 4.3 | 47.0 | 50.5 | 3.5 | **3.08** |

the proportion of FLOP reduction compared to the FLOPs of the dense model. $\Delta$ means the difference in accuracy between the related works and dense model. In particular, the symbol * represents that $\Delta$ is calculated by comparing the performance with the dense counterpart using knowledge distillation (KD). With KD, the Top-1 accuracy for dense DeiT-base and DeiT-small is 83.36% and 81.2%, respectively.

Besides, the details of related works of both tables are outlined in Table 16 for readers' convenience. Among the related works, we would like to provide a detailed comparison of our method with NViT (Yang et al., 2023), which represents the state-of-the-art in DeiT compression. 1) In terms of accuracy, both our method and NViT achieve nearly lossless Top-1 accuracy for sparse DeiT-base and DeiT-small. 2) Regarding FLOPs reduction, our method achieves the highest reduction for both DeiT-base (71.6% ↓) and DeiT-small (54.3% ↓) when compared with NViT and other related works. 3) For parameter reduction, our method achieves the highest reduction for DeiT-small (57.9% ↓) among related works, while NViT achieves the highest reduction for DeiT-base (80.4% ↓), compared to our 73.8% ↓. It is noteworthy that NViT combines multiple strategies for compressing DeiTs, including global structural pruning, 2:4 pruning, and parameter redistribution. In contrast, our work focuses solely on V:N:M sparsity and has the potential to further enhance parameter reduction ratios when combined with other compression strategies. 4) For speedups, both NViT and our method yield significant practical speedups for sparse DeiTs. Specifically, NViT-B with ASP achieves speedups of 1.86x and 1.85x on V100 and RTX 3080 GPUs, respectively, while our 64:2:8-sparse DeiT-base achieves a 2.08x speedup on RTX 3090 GPUs.

Table 16: Details of related works used in Table 5 and 6

| Abbr. | Reference | Conference |
|---|---|---|
| T2T-ViT-24 (Yuan et al., 2021) | Tokens-to-token vit: Training vision Transformers from scratch on imagenet | CVPR2021 |
| PVT (Wang et al., 2021) | Pyramid vision Transformer: A versatile backbone for dense prediction without convolutions | ICCV2021 |
| AutoFormer (Chen et al., 2021a) | Autoformer: Searching Transformers for visual recognition | ICCV2021 |
| S2ViTE (Chen et al., 2021b) | Chasing sparsity in vision Transformers: An end-to-end exploration | NeurIPS2021 |
| Evo-ViT (Xu et al., 2022) | Evo-vit: Slow-fast token evolution for dynamic vision Transformer | AAAI2022 |
| EViT(-DeiT-B) (Liang et al., 2022) | Not all patches are what you need: Expediting vision Transformers via token reorganization | ICLR2022 |
| DynamicViT (Rao et al., 2021) | Dynamicvit: Efficient vision Transformers with dynamic token sparsification | NeurIPS2021 |
| ViT-Slim (Chavan et al., 2022) | Vision Transformer slimming: Multi-dimension searching in continuous optimization space | CVPR2022 |
| VTP (Zhu et al., 2021) | Vision Transformer Pruning | ARXIV2021 |
| PS-ViT (Tang et al., 2022) | Patch slimming for efficient vision Transformers | CVPR2022 |
| UVC (Yu et al., 2022) | Unified visual transformer compression | ICLR2022 |
| AdaViT (Yin et al., 2021) | Adavit: Adaptive tokens for efficient vision Transformer | ARXIV2022 |
| SAViT (Zheng et al., 2022) | SAVIT: Structure aware vision Transformer pruning via collaborative optimization | NeurIPS2022 |
| NViT (Yang et al., 2023) | Global vision Transformer pruning with hessian-aware saliency | ICCV2023 |
| LPViT (Xu et al.) | LSP: Low-Power Semi-structured Pruning for Vision Transformers | ARXIV2024 |

