# OpenReview forum: "Beyond 2:4: Exploring V:N:M Sparsity for Efficient Transformer Inference on GPUs"
_ICLR.cc/2025/Conference — Submitted to ICLR 2025_

### Official Review · Reviewer_QK6w · 2024-10-28

**Soundness:** 3
**Presentation:** 3
**Contribution:** 3
**Rating:** 6
**Confidence:** 4

**Summary:**

This paper studies accuracy and speedup tradeoff of V:N:M sparse Transformers. It introduces a heuristic method for parameter selection, a channel permutation approach for low budget training and a three-stage LoRA training procedure applicable to various scenarios. Experimental results show that the proposed methods significantly accelerate the V:N:M sparse Transformers while maintaining lossless accuracy compared to 2:4 sparsity.

**Strengths:**

1. The authors propose three training approaches to enhance the applicability and accuracy of V:N:M sparse Transformers. They present a heuristic V and M selection method to find the tradeoff between accuracy and speedup for V:N:M sparse Transformers. To improve the accuracy within limited training budgets, they introduce a channel permutation approach. Finally, they introduce a Dense-Dynamic-Fixed mask training procedure to maintain the model accuracy.

2. The paper is well-written and easy to follow up. The introduction and motivation are clear and well-organized. The experiments are comprehensive.

**Weaknesses:**

1. Why is the paper claimed that only a 2:4 sparse pattern can be accelerated by sparse tensor cores? I have some doubts about it. One could write a custom CUDA kernel and accelerate based on the indices of the weights.
2. How did you implement the kernel using sparse tensor core? For higher sparsity (like 2:12, 2:24), the observed speedup falls short of the theoretical speedup, possibly due to inefficiencies in the implementation. Moreover, would you consider using cusparse/cublas or other CUDA kernels to compare speedup with the proposed method at highly sparsity?
3. In Fig.4(a), the authors compare accuracy improvement under similar speedup conditions. In my view, more parameters generally lead to higher accuracy, so it would be fairer to compare accuracy at the same sparsity. Fig.4(a) shows how V and M selection impacts the accuracy and speedup of V:N:M sparse Transformers. However, the V:N:M setting in Fig.8 differs. Does this imply that V and M selection is not applied in Fig.8?

**Questions:**

Please see the above.

---

> ### Author Response · Authors · 2024-11-21
> **(Q1) Thanks for your great questions! Here is our answer to Question 1.**
>
> Thank you very much for your thoughtful review and constructive feedback. We appreciate your recognition of the strengths of our paper and would like to address the weaknesses you raised.
> ___
> Q1: Why is the paper claimed that only a 2:4 sparse pattern can be accelerated by sparse tensor cores? I have some doubts about it. One could write a custom CUDA kernel and accelerate based on the indices of the weights.
>
> **A1**: Thanks for your comment. We would like to clarify our claim by discussing three specific scenarios in which a custom CUDA kernel can accelerate sparse computations based on the weight indices:
>
> 1. **Unstructured sparsity**. When the indices of non-zero weights are completely irregular, a CUDA kernel can only utilize **CUDA core** instructions to perform sparse matrix-matrix multiplications, which allows for the flexible omission of zero operations.
>    - However, it is important to note that the computational power of CUDA cores is significantly lower than that of tensor cores on GPUs. For example, the peak computational power of CUDA cores on an RTX 4090 GPU is **82.6 TFLOPS**, while the peak power of tensor cores (with FP16 accumulation) is **330.3 TFLOPS**. Given that contemporary dense Transformer models predominantly rely on tensor cores for inference [1], unstructurally sparse Transformers using CUDA cores barely achieve superior inference speeds compared to their tensor core counterparts.
>    - Additionally, even when both dense and sparse Transformers execute inference solely on CUDA cores, the irregular computation and memory access patterns associated with unstructured sparsity hinder their ability to maintain competitive inference speeds unless the sparsity exceeds 95% [2]. However, at such high sparsity levels, the accuracy of most Transformers tends to collapse.
> 2. **Block sparsity**. In cases where the non-zero weights are organized into blocks—some of which are dense while others are entirely sparse—block-sparse GEMMs can be effectively accelerated using **dense tensor cores** [2] rather than sparse tensor cores.
> 3. **2:4 and V:N:M sparsity**. When the arrangement of non-zero weight indices adheres to the 2:4 sparse format, i.e., at most two non-zero values among every four consecutive weights, it becomes feasible to utilize **sparse tensor cores** through the "mma.sp()" instruction within a CUDA kernel to accelerate sparse Transformers. Theoretically, sparse tensor cores offer double the computational capacity of dense tensor cores. Similarly, to comply with the 2:4 sparse format, V:N:M sparsity also implements 2:4 pruning for a weight tensor after vector-wise pruning.
>
> [1] [Tensor core usage in CuBLAS.](https://docs.nvidia.com/cuda/cublas/)
>
> [2] [Accelerating Matrix Multiplication with Block Sparse Format and NVIDIA Tensor Cores](https://developer.nvidia.com/blog/accelerating-matrix-multiplication-with-block-sparse-format-and-nvidia-tensor-cores/)
>
> ___

---

> ### Author Response · Authors · 2024-11-21
> **(Q2) Thanks for your great questions! Here is our answer to Question 2.**
>
> Q2: How did you implement the kernel using sparse tensor core? For higher sparsity (like 2:12, 2:24), the observed speedup falls short of the theoretical speedup, possibly due to inefficiencies in the implementation. Moreover, would you consider using cusparse/cublas or other CUDA kernels to compare speedup with the proposed method at highly sparsity?
>
> **A2**: Thank you.
> 1. We utilize the same GPU kernel as employed in the Venom paper to accelerate V:N:M-sparse Transformers. For clarity, we have highlighted key distinctions between our work and the initial V:N:M work in the revised paper. First, as shown in Figure 2, our work and the initial work for V:N:M sparsity lies in distinct phases in the workflow of compressing a Transformer. The initial work primarily proposes GPU acceleration kernels, i.e., Figure 2(b). In contrast, our research emphasizes enhancing the accuracy of V:N:M-sparse Transformers under various scenarios. Our proposed framework within the three novel techniques is depicted in Figure 2(a).
>
>    Besides, here we also would like to detail the significantly distinguishable contributions and advancements our work makes beyond the initial V:N:M sparsity study and other related works:
>    - **our work identifies and addresses critical issues that have long affected the field of Transformer 2:4 sparsity, yet have received insufficient attention until now**. That is, 2:4-sparse Transformers in practice provide very limited end-to-end speedups, while the promising V:N:M sparsity, although capable of delivering higher speedups, has not demonstrated comparable accuracy for sparse Transformers relative to 2:4 sparsity. Our work exactly aims to bridge this gap by proposing a framework with three innovative techniques that algin the accuracy V:N:M sparsity with that of 2:4 sparsity. In doing so, our work significantly contributes to the field by promoting the widespread adoption of V:N:M sparsity as a truly effective solution for compressing Transformers.
>    - **Our work tackles three key fundamental challenges that prevented V:N:M-sparse Transformer from achieving the desired accuracy—challenges that have never been addressed in previous research**. The three challenges are:(1) selecting appropriate values for V and M, (2) enhancing the accuracy of V:N:M sparsity under limited training budgets, and (3) implementing V:N:M-sparse LoRA training for large language models (LLMs). These challenges are critical and cannot be overlooked. Accordingly, our proposed three techniques to address challenges are also novel contributions that are unique to the context of V:N:M sparsity, as elaborated in the main text of our paper. With the proposed three techniques, the novelty of our work also lies in the breadth of scenarios that our framework covers, from visual pretraining to LoRA training and limited-budget training.
>
> 2. The original Venom paper has compared sparse matrix multiplication (MM) using the V:N:M-sparse GPU kernel with dense MM performed using CuBLAS. Their results demonstrate that V:N:M-sparse MM significantly outperforms dense MM at high levels of sparsity. For your reading convenience, we list part of the results below. Please note that in the table, "Model" refers to that the shapes of MM are from that model.
>
> |          Model          | Bert-base | Bert-base | Bert-large | Bert-large |
> |:-----------------------:|:---------:|:---------:|:----------:|:----------:|
> |        Batchsize        |     8     |     16    |      8     |     16     |
> |      Dense(CuBLAS)      |     1     |     1     |      1     |      1     |
> |  64:2:8 (75% sparsity ) |    2.5    |    2.4    |      3     |     3.2    |
> | 64:2:20 (90% sparsity ) |    4.9    |    5.0    |     6.0    |     6.1    |
> ___

---

> ### Author Response · Authors · 2024-11-21
> **(Q3, Q4) Thanks for your great questions! Here are our answers to Question 3 and 4.**
>
> Q3: In Fig.4(a), the authors compare accuracy improvement under similar speedup conditions. In my view, more parameters generally lead to higher accuracy, so it would be fairer to compare accuracy at the same sparsity.
>
> **A3**: Thank you for your concerns. In our paper, we have also presented substantial results comparing the accuracy at the same sparsity levels. We would like to clarify this further as follows:
> 1. **Table 4** indicates that, using training setting (TS) 1 within our proposed framework, the 64:2:5-sparse DeiT-base achieves a significant accuracy improvement through our V:N:M-specific channel permutation, while maintaining the same parameter count.
> 2. **Table 5** shows that under TS2 in our framework, the 64:2:8-sparse DeiT-base has fewer parameters than related works (22.7M vs. 25.4M) while achieving nearly lossless accuracy compared to its dense counterparts.
> 3. **Table 6** further illustrates that with TS2 in our framework, the 64:2:5-sparse Swin-base attains the same classification accuracy as state-of-the-art models but requires significantly fewer parameters (35.9M vs. 64.1M). Additionally, for object detection, the 32:2:6-sparse H-DETR using TS2 achieves a higher mean average precision (mAP) than fixed mask training (45.3% vs. 44.1%) while utilizing the same number of parameters.
> 4. **Table 7** demonstrates that with TS3 in our framework, the 64:2:5-sparse Llama2-7B, despite having fewer parameters than related works employing 2:4 sparsity, achieves better perplexity on downstream tasks.
>
> Altogether, on one hand, we list tables or figures to demonstrate that our proposed framework can lead to lower parameters or FLOPs over related works for a wide range of V:N:M-sparse Transformers while maintaining comparable or higher accuracy, such as Table 5,6 and 7. On the other hand, we would also like to use tables or figures to illustrate the effectiveness of those individual novel techniques embedded within the framework, such as Figure 4(a), which highlights our V and M selection technique.
>
> Besides, we would like to clarify that for V:N:M-sparse Transformers, "more parameters generally lead to higher accuracy" is not necessarily tenable. To illustrate this, we present a specific example in Table 5.1. Among the three (V, M) configurations (16,16), (32,16), and (128,15) for DeiT-base models, the 128:2:15-sparse DeiT-base has the highest parameter count, yet the 16:2:16-sparse DeiT-base, which exhibits the highest MD, achieves the best accuracy, significantly surpassing the other two configurations. The principle behind the example is that, **both the sparse granularity determined by V and the retained parameter counts dictated by M influence the accuracy of V:N:M-sparse Transformers**. Compared to relying solely on parameter counts, MD accounts for both factors, thereby providing a more proper measure of the accuracy of V:N:M-sparse Transformers.
>
> Table 5.1. V:N:M-sparse DeiT-Base 30 epoch LoRA
>    accuracy on downstream tasks
>
> |       (V,M)       |  (16,16) | (32,16) | (128,15) |
> |:-----------------:|:--------:|:-------:|:--------:|
> |     Params.(M)    |   10.8   |   10.8  | **11.5** |
> |   Simplified MD   |   20837  |  18674  |   18168  |
> |   Bird Accu.(%)   |   84.1   |   82.1  |   79.5   |
> | Vehicle Accu.(%)  |   77.4   |   75.2  |   72.6   |
> | CIFAR10 Accu.(%)  |   86.5   |   85.4  |   84.0   |
> | CIFAR100 Accu.(%) |   55.9   |   53.3  |   50.5   |
> |      AVG (%)      | **76.0** |    74   |   71.7   |
>
> ___
> Q4: Fig.4(a) shows how V and M selection impacts the accuracy and speedup of V:N:M sparse Transformers. However, the V:N:M setting in Fig.8 differs. Does this imply that V and M selection is not applied in Fig.8?
>
> **A4**: Yes. Figure 8 in Appendix D of our paper is to detail the speedups achieved by various ViT models under different V:N:M sparsity. It is noteworthy that Figure 8 is not intended to demonstrate the effectiveness of our V and M selection technique. From this question, we sincerely appreciate your careful review of our paper and value your attention to detail.

---

> ### Comment · Reviewer_QK6w · 2024-11-26
>
> Thanks for your response, which addressed my concern.
> After reading my and others' responses, I will increase my rate.

---

> > ### Author Response · Authors · 2024-11-28
> > **Thank you very much for your recognition!**
> >
> > Dear Reviewer QK6w,
> >
> > Thank you for your decision! We sincerely appreciate your recognition of our work. We believe your support not only motivates us but also significantly advances the field of Transformer compression and the technology of sparse inference.
> >
> > Thank you once again for your valuable feedback.
> >
> > Warm regards,
> >
> > authors

---

### Official Review · Reviewer_Dgv5 · 2024-11-02

**Soundness:** 3
**Presentation:** 2
**Contribution:** 2
**Rating:** 5
**Confidence:** 3

**Summary:**

The conventional 2:4 sparsity has been the only sparse pattern capable of accelerating GPU sparse tensor cores, and its performance was well-established.
However, the paper claims that 2:4 sparsity has limitations, such as low actual speedups and support for only fixed sparsity ratios.
To address this, the paper adopts the V:N:M approach.
V:N:M sparsity combines n:m pruning with structured pruning and claims to overcome the fixed 50% pruning ratio limitation of traditional 2:4 sparsity.

**Strengths:**

1. Figure 1 is well-drawn and easy to understand.
2. The experiments conducted on the ViT model and LLM model are excellent.

**Weaknesses:**

1. [Novelty] I am not sure about the differences between this paper and the Venom paper [1]. Wasn't V:N:M proposed in the Venom paper [1]? How does it differ from Venom [1]?
2. The Venom paper proposed a GPU kernel for speedup sparse model on GPUs. how does this paper handle processing on the GPU? Does it propose a dedicated GPU kernel? If so, what are the differences compared to the Venom [1] GPU kernel?
3. The title of the paper includes "Transformer"—what specific characteristics of Transformers does it take into account?
4. Looking at Table 5, the caption lacks sufficient explanation. For instance, a clear description of "$\Delta$" is necessary in the caption.


[1] Venom: A vectorized N:M format for unleashing the power of sparse tensor cores. SC 2023

**Questions:**

See weakness.

---

> ### Author Response · Authors · 2024-11-21
> **(Q1) Thanks for your positive feedback and valuable comments! Here is our answer to question 1.**
>
> Q1: [Novelty] Differences between this paper and the Venom paper [1]. Wasn't V:N:M proposed in the Venom paper ? How does it differ from Venom ?
>
> **A1**: Thanks. we would like to clarify that the concept of "V:N:M sparsity" was first introduced in the Venom paper; however, that initial work concentrated on the GPU kernel design for V:N:M-sparse GEMMs. In contrast, our research focuses on improving the accuracy of a wide range of V:N:M-sparse Transformers under different scenarios.
> Besides, we would like to detail the significantly distinguishable contributions and advancements our work makes beyond the initial V:N:M sparsity study and other related works:
> 1. **Our work identifies and addresses critical issues that have long affected the field of Transformer 2:4 sparsity, yet have received insufficient attention until now**. That is, 2:4-sparse Transformers in practice provide very limited end-to-end speedups, while the promising V:N:M sparsity, although capable of delivering higher speedups, has not demonstrated comparable accuracy for sparse Transformers relative to 2:4 sparsity. Our work exactly aims to bridge this gap by proposing a framework with three innovative techniques that algin the accuracy V:N:M sparsity with that of 2:4 sparsity. In doing so, our work significantly contributes to the field by promoting the widespread adoption of V:N:M sparsity as a truly effective solution for compressing Transformers.
> 2. **Our work tackles three key fundamental challenges that prevented V:N:M-sparse Transformer from achieving the desired accuracy—challenges that have never been addressed in previous research**. The three challenges are:(1) selecting appropriate values for V and M, (2) enhancing the accuracy of V:N:M sparsity under limited training budgets, and (3) implementing V:N:M-sparse LoRA training for large language models (LLMs). These challenges are critical and cannot be overlooked. Accordingly, our proposed three techniques to address challenges are also novel contributions that are unique to the context of V:N:M sparsity, as elaborated in the main text of our paper. With the proposed three techniques, the novelty of our work also lies in the breadth of scenarios that our framework covers, from visual pretraining to LoRA training and limited-budget training.
> 3. We further clarify the originality in our paper. we have highlighted key distinctions between our work and the initial V:N:M work in the paper. First, as shown in Figure 2, our work and the initial work for V:N:M sparsity lies in distinct phases in the workflow of compressing a Transformer. The initial work primarily proposes GPU acceleration kernels, i.e., Figure 2(b). In contrast, our research emphasizes enhancing the accuracy of V:N:M-sparse Transformers under various scenarios. Our proposed framework within the three novel techniques is depicted in Figure 2(a). Importantly, combining the three novel techniques with other factors, such as pruning criteria, to form our comprehensive framework is non-trivial. For instance, under the limited training budget, employing RIA-based pruning rather than ABS-based pruning is essential for achieving better accuracy, as demonstrated below.
>
> Table 4.1 Results of 64:2:5-sparse DeiT-base on downstream tasks using RIA and ABS-based pruning, respectively.
> | Downstream tasks | BIRD  | BIRD  | BIRD  | VEHICLE | VEHICLE | VEHICLE | CIFAR-10 | CIFAR-10 | CIFAR-10 | CIFAR-100 | CIFAR-100 | CIFAR-100 | Average |
> | ---------------- | ----- | ----- | ----- | -------- | -------- | -------- | -------- | -------- | -------- | --------- | --------- | --------- | -------- |
> | Dense model(%)   | 97.8  | 97.8  | 97.8  | 97.3     | 97.3     | 97.3     | 98.1     | 98.1     | 98.1     | 87.54     | 87.54     | 87.54     | Average  |
> | Permutation Method | Importance Score | No    | Ours  | △     | No       | Ours     | △       | No       | Ours     | △       | No        | Ours      | △       | Average  |
> | Upon Pruning(%)  |   RIA    | 81.6  | 87.6  | 6.0    | 78.6     | 85.7     | 7.1      | 84.42    | 89.85    | 5.43     | 57.65     | 63.07     | **5.42**     | 5.87     |
> | 30 epochs-all params(%) | RIA   | 96.5  | 96.9  | 0.4    | 95.7     | 96.4     | 0.7      | 97.51    | 97.73    | 0.22     | 85.54     | 86.65     | **1.11**     | 0.71     |
> | 30 epochs-LoRA(%) |    RIA   | 96.3  | 96.9  | 0.6    | 94.7     | 96.4     | 1.7      | 97.26    | 97.73    | 0.47     | 83.96     | 86.65     | **2.69**     | 1.63     |
> | Upon Pruning(%)  |    ABS   | 41.2  | 43.1  | 1.9    | 29.4     | 43.4     | 14.0     | 37.09    | 35.40    | -1.69    | 10.70     | 15.01     | 4.31     | 4.63     |
> | 30 epochs-all params(%) | ABS   | 96.0  | 95.9  | -0.1   | 95.4     | 95.3     | -0.1     | 97.39    | 97.50    | 0.11     | 84.83     | 85.63     | 0.80     | 0.18     |
> | 30 epochs-LoRA(%) |   ABS    | 95.9  | 95.8  | -0.1   | 94.1     | 95.4     | 1.3      | 96.68    | 97.03    | 0.35     | 82.80     | 83.35     | 0.55     | 0.52     |

---

> ### Author Response · Authors · 2024-11-21
> **(Q2,Q3,Q4) Thanks for your positive feedback and valuable comments! Here are our answers to question 2,3,4.**
>
> Q2: The Venom paper proposed a GPU kernel for speedup sparse models on GPUs. how does this paper handle processing on the GPU? Does it propose a dedicated GPU kernel? If so, what are the differences compared to the Venom [1] GPU kernel?
>
> **A2**: Thank you. We utilize the same GPU kernel as employed in the Venom paper to accelerate V:N:M-sparse Transformers. Our major contribution is the enhancement of accuracy and applicability of V:N:M-sparse Transformers, as stated before.
> ___
> Q3: The title of the paper includes "Transformer"—what specific characteristics of Transformers does it take into account?
>
> **A3**: Thanks. Given that the original V:N:M GPU kernel in the Venom paper only provides speedup for linear layers, we selected Transformers which include a number of linear layers in their structure for employing V:N:M sparsity to obtain high end-to-end speedups. However, your feedback is greatly appreciated and we put the development of V:N:M-sparse GPU kernels for convolutional operations as our future work.
> ___
> Q4: Looking at Table 5, the caption lacks sufficient explanation. For instance, a clear description of " $\Delta$
> " is necessary in the caption.
>
> **A4**: Thank you very much for your carefulness.
> 1. In our revised paper, we have linked the caption with the description of $\Delta$ and $\downarrow \%$. For instance, in the case of the DeiT-base model, as shown below, the $\Delta$ means the difference in accuracy between the related works and the dense DeiT-base. The first $\downarrow \%$ indicates the proportion of parameter reduction relative to the parameters of the dense DeiT-base, while the second $\downarrow \%$ signifies the proportion of FLOP reduction compared to the FLOPs of the dense DeiT-base.
>
> |     Model     | Params.(M) |          **$\downarrow$%**       | FLOPs(G) |          **$\downarrow$%**       | Top1-acc. (\%) | **$\Delta$** |
> |:-------------:|:----------:|:--------------------------------:|:--------:|:--------------------------------:|:--------------:|:------------:|
> |     DeiT-B    |    86.6    |                 -                |   17.6   |                 -                |      81.84     |       -      |
> | Related works |      $A$    | $\frac{86.6-A}{86.6}\times 100%$ |     $B$    | $\frac{17.6-B}{17.6}\times 100%$ |        $C$       |   $C-81.84$  |
> |      ...      |     ...    |                ...               |    ...   |                ...               |       ...      |      ...     |
>
> 2. Furthermore, in Appendix J of our revised paper, we outline the references corresponding to each related work in Table 5 to enhance clarity. For your reading convenience, we also present the list here:
>
> Table 4.2 Details of related works in Table 5 of the main paper
> | Abbr. name    | Reference                                                                                  | Conference  |
> |---------------|--------------------------------------------------------------------------------------------|-------------|
> | T2T-ViT-24    | Tokens-to-token vit: Training vision Transformers from scratch on imagenet                 | CVPR2021    |
> | PVT           | Pyramid vision Transformer: A versatile backbone for dense prediction without convolutions | ICCV2021    |
> | AutoFormer    | Autoformer: Searching Transformers for visual recognition                                  | ICCV2021    |
> | S2ViTE        | Chasing sparsity in vision Transformers: An end-to-end exploration                         | NeurIPS2021 |
> | Evo-ViT       | Evo-vit: Slow-fast token evolution for dynamic vision Transformer                          | AAAI2022    |
> | EViT(-DeiT-B) | Not all patches are what you need: Expediting vision Transformers via token reorganization | ICLR2022    |
> | DynamicViT    | Dynamicvit: Efficient vision Transformers with dynamic token sparsification                | NeurIPS2021 |
> | ViT-Slim      | Vision Transformer slimming: Multi-dimension searching in continuous optimization space    | CVPR2022    |
> | VTP           | Vision Transformer Pruning                                                                 | ARXIV2021   |
> | PS-ViT        | Patch slimming for efficient vision Transformers                                           | CVPR2022    |
> | UVC           | Unified visual transformer compression                                                     | ICLR2022    |
> | AdaViT        | Adavit: Adaptive tokens for efficient vision Transformer                                   | ARXIV2022   |
> | SAViT         | SAVIT: Structure aware vision Transformer pruning via collaborative optimization           | NeurIPS2022 |
> | NViT          | Global vision Transformer pruning with hessian-aware saliency                              | ICCV2023    |
>
> 3. Besides, we conducted a thorough review of all tables and figures to ensure clarity and completeness throughout the paper. We sincerely appreciate your valuable feedback.

---

> ### Comment · Reviewer_Dgv5 · 2024-11-26
> **Thanks**
>
> Thanks the authors for providing the feedback. By checking the rebuttal and comments, i think there are still 'lack of novelty' of this paper. So i maintain my score.

---

> > ### Author Response · Authors · 2024-12-01
> > **(Part 2) Thanks for your feedback! Hope for more discussions.**
> >
> > 3. **Broader impact**. We are excited to report that, following our recent efforts, we have successfully demonstrated that the proposed novel methods also perform effectively with ResMLP networks, which utilize linear layers for both channel and patch fusion. As shown in Table 4.3, our V:N:M-specific channel permutation enhances the accuracy of 64:2:5-sparse ResMLP_S36 under limited training budgets. Furthermore, Figure 4.1 illustrates that our V and M selection method enables V:N:M-sparse ResMLP_B24 to achieve improved accuracy-speedup trade-offs. The success on V:N:M-sparse ResMLP networks signifies a broader impact that our proposed methods prove valuable and effective for network architectures where linear layers play predominate roles.
> >
> > [Figure 4.1 Average Top-1 accuracy of ResMLP_B24 on downstream tasks with different V and M vlaues](https://anonymous.4open.science/r/conf_dat_Anonymous-8055/downstream_resmlp-b24.pdf)
> >
> > Table 4.3 Results of 64:2:5 ResMLP_S36 on downstream tasks under limited training budgets
> > | Downstream tasks | BIRD | BIRD | BIRD | VEHICLE | VEHICLE | VEHICLE | CIFAR-10 | CIFAR-10 | CIFAR-10 | CIFAR-100 | CIFAR-100 | CIFAR-100 | Average |
> > | --- | --- | --- | --- | --- | --- | --- | --- | --- | --- | --- | --- | --- | --- |
> > | Dense model(%) | 97.5 | 97.5 | 97.5 | 96.7 | 96.7 | 96.7 | 97.4 | 97.4 | 97.4 | 84.5 | 84.5 | 84.5 | Average |
> > | Permutation Method | No | Ours | △ | No | Ours | △ | No | Ours | △ | No | Ours | △ | - |
> > | 30 epochs-all params.(%) | 96.5 | 96.7 | 0.2 | 96.1 | 96.4 | 0.3 | 97.1 | 97.2 | 0.1 | 83.9 | 84.5 | 0.6 | **0.3** |
> > | 30 epochs-LoRA(%) | 95.4 | 95.7 | 0.3 | 93.7 | 95.1 | 1.4 | 94.5 | 95.2 | 0.7 | 78.3 | 79.8 | 1.5 | **1.0** |
> > | Training-free Pruning(%) | 66.4 | 87.9 | 21.5 | 66.9 | 80.7 | 13.8 | 63.1 | 78.6 | 15.5 | 40.8 | 51.9 | 11.1 | **15.5** |
> >
> > We would be grateful if you could take the time to review these clarifications and updates, and provide a re-evaluation of our work. We sincerely hope to receive your further feedback and will spare no effort to address any remaining concerns you may have.

---

> > ### Author Response · Authors · 2024-12-02
> > **Request for feedback from Reviewer Dgv5**
> >
> > Dear reviewer Dgv5,
> >
> > We sincerely appreciate your time and previous feedback. As the deadline for the Author/Reviewer interactive discussion approaches, please let us know if you need any further clarifications or details. We humbly and respectfully request your reconsideration of our work in light of the technical novelties and contributions we have previously outlined. Your expertise is invaluable, and we are eager to address any remaining concerns. Thank you very much for your reconsideration.
> >
> > Best regards,
> >
> > authors

---

> ### Author Response · Authors · 2024-12-01
> **(Part 1) Thanks for your feedback! Hope for more discussions.**
>
> Thank you very much for your feedback. We humbly and respectfully request your renewed attention to our technical novelty and contributions:
>
> 1. Our work primarily focuses on algorithmic advancements rather than GPU kernel acceleration. The novelty in our algorithms is noteworthy in both scope and effectiveness. We propose **three novel methods for V:N:M-sparse Transformers: V and M selection, V:N:M-specific channel permutation, and three-staged LoRA training method**. In particular:
>    - For our proposed V and M selection method, we **introduce the concept of mask diversity (MD) for V:N:M Transformers for the first time and provide theoretical proof demonstrating that the MD of different V and M configurations can be efficiently compared using relative MD**. This relative MD allows for the rapid selection of appropriate V and M values, enabling the corresponding V:N:M-sparse Transformer to achieve optimal accuracy and speedup tradeoffs. The effectiveness of our V and M selection method is thoroughly validated across all training settings, as shown in Figures 4 and 10 of our paper, and across multiple Transformer architectures, as illustrated in [Figures 11](https://anonymous.4open.science/r/conf_dat_Anonymous-8055/downstream_vit-large.pdf) (ViT-large), [12](https://anonymous.4open.science/r/conf_dat_Anonymous-8055/downstream_vit-huge.pdf) (ViT-huge), and [13](https://anonymous.4open.science/r/conf_dat_Anonymous-8055/downstream_Llama2-13B.pdf) (Llama2-13B) of our paper.
>    - For our V:N:M-specific channel permutation method, we **provide insights into how both input and output channel permutations of weight matrices can influence the accuracy of V:N:M-sparse Transformers—an aspect that differs significantly from 2:4 sparsity**. We model the V:N:M-specific channel permutation as an alternative optimization problem to efficiently solve for the permutation matrices. The effectiveness of our proposed channel permutation method is extensively validated, as demonstrated in Tables 12, 13, 14, and 15 in our paper.
>
>    - In our three-staged LoRA training method for  V:N:M-sparse LLMs, we **propose, for the first time, that a brief duration of dynamic mask changes following dense LoRA initialization can significantly enhance the effectiveness of LoRA training**. This approach optimizes the trade-offs between exploration and exploitation in mask selection. The effectiveness of our three-staged LoRA training method is validated through Table 8 and Figure 5 in our paper.
>
> 2. Our innovations help address three critical challenges that **have never been investigated and addressed by previous research**: the selection of appropriate V and M values for V:N:M-sparse Transformers, the enhancement of accuracy for V:N:M-sparse Transformers under limited training budgets, and the integration of V:N:M sparsity into LoRA to facilitate the fine-tuning of V:N:M-sparse LLMs. More importantly, through these innovations, our framework achieves that **the accuracy of a wide range of V:N:M-sparse Transformers is comparable to or exceeds that of 2:4-sparse Transformers**. Given that V:N:M-sparse Transformers also offer enhanced speedup compared to their dense counterparts and exhibit superior speedup-accuracy scalability, our work represents a significant advancement in establishing V:N:M sparsity as atruly viable alternative to 2:4 sparsity for compressing highly redundant Transformers.

---

> > ### Author Response · Authors · 2024-12-02
> > **Request for feedback from Reviewer Dgv5: Final Day of Reviewer-Author Discussion**
> >
> > Dear Reviewer Dgv5,
> >
> > Thank you for your dedicated time and attention to reviewing our paper. As the discussion stage is coming to an end, please let us know whether we have addressed your concerns or if any further discussions are needed.
> >
> > We look forward to your feedback and sincerely request that you reconsider the technical novelty and contributions of our work.
> >
> > Thank you once again for your valuable contribution to our work.
> >
> > Best regards,
> > authors

---

### Official Review · Reviewer_WKMR · 2024-11-03

**Soundness:** 2
**Presentation:** 3
**Contribution:** 2
**Rating:** 6
**Confidence:** 3

**Summary:**

The paper addresses the limitations of 2:4 sparsity, by thoroughly investigating the application of V:N:M sparsity in vision models and LLMs across multiple tasks. The authors propose methods for selecting optimal V and M values and introduce techniques like V:N channel permutation and a three-stage LoRA training method, designed to enhance the accuracy and applicability of V:N Transformers, particularly in vision and language models.

**Strengths:**

1. The paper goes beyond theoretical contributions by proposing practical methods to improve the deployment of V:N:M
sparsity. Techniques like heuristic selection of V and M values, channel permutation, and three-staged LoRA training are thoughtfully designed to optimize accuracy-speedup trade-offs, even with limited training resources.
2. The authors validate their approach across multiple benchmarks, including vision Transformers and large language models (LLMs), demonstrating that V:N:M sparsity can achieve high speedups with minimal accuracy loss.

**Weaknesses:**

1. The heuristic method for selecting V and M values may not work for extensive Transformers models. My concerns are that even in the paper the proposed methods can achieve lossless performance on specific model DeiT-base and LLama2-7B. But what are the expected results on other models?
2. The novelty and originality may not be enough. This paper is more like an engineering implementation technique. Authors may need to explain originality more.

**Questions:**

1. The heuristic method for selecting V and M values may not work for extensive Transformers models. My concerns are that even in the paper the proposed methods can achieve lossless performance on specific model DeiT-base and LLama2-7B. But what are the expected results on other models?
2. The novelty and originality may not be enough. This paper is more like an engineering implementation technique. Authors may need to explain originality more.
3. "V:N:M-specific channel permutation, which improves the accuracy of V:N:M-sparse Transformers". Why channel permutation can improve accuracy?
4. "a higher MD leads to better Transformer accuracy" Is this claim from extensive experiments or other places?

---

> ### Author Response · Authors · 2024-11-23
> **(Q1) Thank you for your recognition and valuable comments! Here is our answer to question 1.**
>
> Q1：The heuristic method for selecting V and M values may not work for extensive Transformers models.
>
> **A1**: Thank you for your kind concerns. We have extended our proposed V and M selection method to three additional representative Transformer models: ViT-large, ViT-huge, and Llama2-13B. The experimental results demonstrate that V:N:M-sparse Transformers utilizing our selection method consistently achieve superior accuracy-speedup trade-offs compared to those that do not employ this approach. It is clear that our V and M selection method can better scale the V:N:M-sparse Transformers.
>
> Please refer to the figures below for detailed results:
>
> [Figure 3.1 Average Top-1 accuracy of ViT-large on downstream tasks with different V and M vlaues](https://anonymous.4open.science/r/conf_dat_Anonymous-8055/downstream_vit-large.pdf)
>
> [Figure 3.2 Average Top-1 accuracy of ViT-huge on downstream tasks with different V and M vlaues](https://anonymous.4open.science/r/conf_dat_Anonymous-8055/downstream_vit-huge.pdf)
>
> [Figure 3.3 Average scores of Llama2-13B on 5-shot tasks with different V and M values.](https://anonymous.4open.science/r/conf_dat_Anonymous-8055/downstream_Llama2-13B.pdf)

---

> ### Author Response · Authors · 2024-11-23
> **(Q2) Thank you for your recognition and valuable comments! Here is our answer to question 2.**
>
> Q2: The novelty and originality may not be enough. Authors may need to explain originality more.
>
> **A2**: Thanks. We understand your concern about the novelty and originality of our work. We would like to emphasize the significantly distinguishable contributions and advancements our work makes beyond the initial V:N:M sparsity study and other related works.
> 1. **our work identifies and addresses critical issues that have long affected the field of Transformer 2:4 sparsity, yet have received insufficient attention until now**. That is, 2:4-sparse Transformers in practice provide very limited end-to-end speedups, while the promising V:N:M sparsity, although capable of delivering higher speedups, has not demonstrated comparable accuracy for sparse Transformers relative to 2:4 sparsity. Our work exactly aims to bridge this gap by proposing a framework with three innovative techniques that align the accuracy V:N:M sparsity with that of 2:4 sparsity. In doing so, our work significantly contributes to the field by promoting the widespread adoption of V:N:M sparsity as a truly effective solution for compressing Transformers.
> 2. Our work does not simply present an engineering implementation, but rather **tackles three key fundamental challenges that prevented V:N:M-sparse Transformer from achieving the desired accuracy—challenges that have never been addressed in previous research**. The three challenges are:(1) selecting appropriate values for V and M, (2) enhancing the accuracy of V:N:M sparsity under limited training budgets, and (3) implementing V:N:M-sparse LoRA training for large language models (LLMs). These challenges are critical and cannot be overlooked. Accordingly, our proposed three techniques to address challenges are also novel contributions that are unique to the context of V:N:M sparsity, as elaborated in the main text of our paper. With the proposed three techniques, the novelty of our work also lies in the breadth of scenarios that our framework covers, from visual pretraining to LoRA training and limited-budget training.
> 3. We further clarify the originality in our paper. we have detailed and highlighted key distinctions between our work and the initial V:N:M work in the paper. First, as shown in Figure 2, our work and the initial work for V:N:M sparsity lies in distinct aspects for compressing a Transformer. The initial work primarily proposes GPU acceleration kernels for V:N:M-sparse general matrix multiply (GEMM) operations, i.e., Figure 2(b). In contrast, our research emphasizes enhancing the accuracy of V:N:M-sparse Transformers under various scenarios. Our proposed framework within the three novel techniques is depicted in Figure 2(a). Importantly, combining the three novel techniques with other factors, such as pruning criteria, to form our comprehensive framework is non-trivial. For instance, under the limited training budget, employing RIA-based pruning rather than ABS-based pruning is essential for achieving better accuracy, as demonstrated below.
>
> Table 3.1 Results of 64:2:5-sparse DeiT-base on downstream tasks using RIA and ABS-based pruning, respectively.
> | Downstream tasks | BIRD  | BIRD  | BIRD  | VEHICLE | VEHICLE | VEHICLE | CIFAR-10 | CIFAR-10 | CIFAR-10 | CIFAR-100 | CIFAR-100 | CIFAR-100 | Average |
> | ---------------- | ----- | ----- | ----- | -------- | -------- | -------- | -------- | -------- | -------- | --------- | --------- | --------- | -------- |
> | Dense model(%)   | 97.8  | 97.8  | 97.8  | 97.3     | 97.3     | 97.3     | 98.1     | 98.1     | 98.1     | 87.54     | 87.54     | 87.54     | Average  |
> | Permutation Method | Importance Score | No    | Ours  | △     | No       | Ours     | △       | No       | Ours     | △       | No        | Ours      | △       | Average  |
> | Upon Pruning(%)  |   RIA    | 81.6  | 87.6  | 6.0    | 78.6     | 85.7     | 7.1      | 84.42    | 89.85    | 5.43     | 57.65     | 63.07     | **5.42**     | 5.87     |
> | 30 epochs-all params(%) | RIA   | 96.5  | 96.9  | 0.4    | 95.7     | 96.4     | 0.7      | 97.51    | 97.73    | 0.22     | 85.54     | 86.65     | **1.11**     | 0.71     |
> | 30 epochs-LoRA(%) |    RIA   | 96.3  | 96.9  | 0.6    | 94.7     | 96.4     | 1.7      | 97.26    | 97.73    | 0.47     | 83.96     | 86.65     | **2.69**     | 1.63     |
> | Upon Pruning(%)  |    ABS   | 41.2  | 43.1  | 1.9    | 29.4     | 43.4     | 14.0     | 37.09    | 35.40    | -1.69    | 10.70     | 15.01     | 4.31     | 4.63     |
> | 30 epochs-all params(%) | ABS   | 96.0  | 95.9  | -0.1   | 95.4     | 95.3     | -0.1     | 97.39    | 97.50    | 0.11     | 84.83     | 85.63     | 0.80     | 0.18     |
> | 30 epochs-LoRA(%) |   ABS    | 95.9  | 95.8  | -0.1   | 94.1     | 95.4     | 1.3      | 96.68    | 97.03    | 0.35     | 82.80     | 83.35     | 0.55     | 0.52     |

---

> ### Author Response · Authors · 2024-11-23
> **(Q3) Thank you for your recognition and valuable comments! Here is our answer to question 3.**
>
> Q3："V:N:M-specific channel permutation, which improves the accuracy of V:N:M-sparse Transformers". Why channel permutation can improve accuracy?
>
> **A3**: Thank you for your insightful question regarding the function of channel permutation.
> 1) Channel permutation (CP) is based on a property of neural networks: parameter space symmetries [1]. That is, different parameter configurations can yield identical loss values for a neural network. This loss-invariant transformation to a neural network is also called teleportation [2]. CP is exactly a special form of teleportation.
> 2) CP can change the training dynamics [3]. In our work, pruning after conducting V:N:M-specific CP can lead to less loss in weights, which means more important weights, like outlier weights [4] are probably retained. Hence, with our V:N:M-specific CP, the V:N:M-sparse Transformers upon pruning usually possess higher accuracy, which also benefits the subsequential training. Notably, CP is especially essential in scenarios given limited training budgets.
> 3) To further illustrate the effectiveness of our V:N:M-specific CP, we present additional results regarding CP performance under various V:N:M ratios in the limited-training-budget scenario, as shown in Tables 3.2, 3.3, and 3.4 below. The results clearly indicate that our CP method significantly enhances the accuracy of these V:N:M-sparse Transformers.
>
> Table 3.2 Results of 64:2:16 DeiT-base on downstream tasks with 30 training epochs
> | Downstream tasks | BIRD | BIRD | BIRD | VEHICLE | VEHICLE | VEHICLE | CIFAR-10 | CIFAR-10 | CIFAR-10 | CIFAR-100 | CIFAR-100 | CIFAR-100 | Average |
> | --- | --- | --- | --- | --- | --- | --- | --- | --- | --- | --- | --- | --- | --- |
> | Dense model(%) | 97.7 | 97.7 | 97.7 | 97.3 | 97.3 | 97.3 | 98.1 | 98.1 | 98.1 | 87.54 | 87.54 | 87.54 | |
> | Permutation Method | No | Ours | △ | No | Ours | △ | No | Ours | △ | No | Ours | △ |  |
> | Upon Pruning(%) | 7.40 | 8.10 | 0.6 | 5.00 | 7.00 | 2.0 | 10.8 | 12.4 | 1.6 | 1.00 | 1.35 | 0.35 | **1.14** |
> | 30 epochs-all params.(%) | 84.9 | 89.9 | 5.0 | 77.8 | 83.5 | 5.7 | 89.9 | 92.7 | 2.8 | 64.4 | 70.7 | 6.3 | **4.95** |
> | 30 epochs-LoRA(%) | 79.7 | 84.1 | 4.3 | 74.3 | 77.4 | 3.1 | 82.9 | 86.5 | 3.6 | 50.2 | 55.9 | 5.7 | **4.18** |
>
> Table 3.3 Results of 32:2:16 DeiT-base on downstream tasks with 30 training epochs
> | Downstream tasks | BIRD | BIRD | BIRD | VEHICLE | VEHICLE | VEHICLE | CIFAR-10 | CIFAR-10 | CIFAR-10 | CIFAR-100 | CIFAR-100 | CIFAR-100 | Average |
> | --- | --- | --- | --- | --- | --- | --- | --- | --- | --- | --- | --- | --- | --- |
> | Dense model(%) | 97.7 | 97.7 | 97.7 | 97.3 | 97.3 | 97.3 | 98.1 | 98.1 | 98.1 | 87.54 | 87.54 | 87.54 |  |
> | Permutation Method | No | Ours | △ | No | Ours | △ | No | Ours | △ | No | Ours | △ |  |
> | Upon Pruning(%) | 8.10 | 6.10 | -2.0 | 6.60 | 6.20 | -0.4 | 9.98 | 13.6 | 3.6 | 1.24 | 1.21 | -0.03 | **0.29** |
> | 30 epochs-all params.(%) | 83.1 | 87.8 | 4.7 | 75.0 | 81.2 | 6.2 | 88.1 | 91.6 | 3.5 | 61.2 | 67.4 | 6.2 | **5.15** |
> | 30 epochs-LoRA(%) | 78.8 | 82.1 | 3.3 | 72.9 | 75.2 | 2.3 | 81.2 | 85.4 | 4.2 | 48.2 | 53.3 | 5.1 | **3.73** |
>
> Table 3.4 Results of 128:2:15 DeiT-base on downstream tasks with 30 training epochs
> | Downstream tasks | BIRD | BIRD | BIRD | VEHICLE | VEHICLE | VEHICLE | CIFAR-10 | CIFAR-10 | CIFAR-10 | CIFAR-100 | CIFAR-100 | CIFAR-100 | Average |
> | --- | --- | --- | --- | --- | --- | --- | --- | --- | --- | --- | --- | --- | --- |
> | Dense model(%) | 97.7 | 97.7 | 97.7 | 97.3 | 97.3 | 97.3 | 98.1 | 98.1 | 98.1 | 87.54 | 87.54 | 87.54 |  |
> | Permutation Method | No | Ours | △ | No | Ours | △ | No | Ours | △ | No | Ours | △ |  |
> | Upon Pruning(%) | 5.00 | 4.50 | -0.5 | 6.60 | 7.10 | 0.5 | 10.2 | 12.4 | 2.2 | 0.97 | 1.32 | 0.35 | **0.64** |
> | 30 epochs-all params.(%) | 83.2 | 86.0 | 2.8 | 74.0 | 78.8 | 4.8 | 88.1 | 91.0 | 2.9 | 60.8 | 65.6 | 4.8 | **3.83** |
> | 30 epochs-LoRA(%) | 77.2 | 79.5 | 2.3 | 70.4 | 72.6 | 2.2 | 79.7 | 84.0 | 4.3 | 47.0 | 50.5 | 3.5 | **3.08** |
>
> [1] Improving convergence and generalization using parameter symmetries. ICLR 2024
>
> [2] Neural Teleportation. Mathematics 2023, 11(2), 480
>
> [3] Level Set Teleportation: An Optimization Perspective. Arxiv 2024.
>
> [4] Outlier weighed layerwise sparsity (owl): A missing secret sauce for pruning llms to high sparsity. ICML 2024.

---

> ### Author Response · Authors · 2024-11-23
> **(Q4) Thank you for your recognition and valuable comments! Here is our answer to question 4.**
>
> Q4："a higher MD leads to better Transformer accuracy" Is this claim from extensive experiments or other places?
>
> **A4**: Thank you for your question regarding the relation between MD and Transformer accuracy.
> 1. The original opinion that the structure constraints of sparse patterns cause accuracy degradation is from the paper [5], where the authors employ mask diversity (MD) to quantify the structure constraints of a sparse pattern. They have validated in ResNet models that a higher MD leads to better accuracy.
> 2. However, [5] does not give the MD of the V:N:M sparse pattern, nor does it employ MD in Transformers, which is non-trivial. Instead, we propose Formula 5 in the paper to determine MD for V:N:M-sparse Transformers. Furthermore, we prove that for the same Transformer, the relative order of MD of different (V,M) configurations is irrespective of the Transformer weight shapes. This finding largely simplifies the computation of MD of V:N:M-sparse Transformer, which facilitates the efficient execution of the proposed V and M selection techniques.
> 3. Our experiments empirically show that a higher MD leads to better accuracy also holds for V:N:M-sparse Transformers. In particular, besides the results in the main paper, we also present a concrete example as Table 3.5 below. Among three (V,M) configurations, although 128:2:15-sparse DeiT-base has the most parameters, the 16:2:16-sparse DeiT-Base which has the highest MD also has the highest accuracy with a significant gap beyond the other two V:N:M-sparse Transformers.
>
> Table 3.5. V:N:M-sparse DeiT-Base 30 epoch LoRA
>    accuracy on downstream tasks
>
> |       (V,M)       |  (16,16) | (32,16) | (128,15) |
> |:-----------------:|:--------:|:-------:|:--------:|
> |     Params.(M)    |   10.8   |   10.8  | **11.5** |
> |   Simplified MD   |   20837  |  18674  |   18168  |
> |   Bird Accu.(%)   |   84.1   |   82.1  |   79.5   |
> | Vehicle Accu.(%)  |   77.4   |   75.2  |   72.6   |
> | CIFAR10 Accu.(%)  |   86.5   |   85.4  |   84.0   |
> | CIFAR100 Accu.(%) |   55.9   |   53.3  |   50.5   |
> |      AVG (%)      | **76.0** |    74   |   71.7   |
>
> [5] Accelerated sparse neural training: A provable and efficient method to find n: m transposable masks. NeurIPS 2021.

---

> > ### Comment · Reviewer_WKMR · 2024-11-25
> > **reviewer's response**
> >
> > Hello authors,
> >    Thanks a lot for the detailed response. The response addressed my concern. I will increase my rate.

---

> > > ### Author Response · Authors · 2024-11-26
> > > **Thank you so much for your recognition!**
> > >
> > > Dear reviewer WKMR,
> > >
> > > Thank you so much for your encouraging feedback! We are thrilled to hear that our detailed response addressed your concerns. Your decision to increase your rating undoubtedly means a great deal to us.
> > >
> > > We believe our research represents a crucial step toward promoting the widespread adoption of V:N:M sparsity as a truly effective solution for compressing Transformers. Thank you once again for your invaluable recognition.
> > >
> > >
> > > Warm regards,
> > >
> > > authors

---

### Official Review · Reviewer_W4Dx · 2024-11-04

**Soundness:** 3
**Presentation:** 3
**Contribution:** 3
**Rating:** 6
**Confidence:** 4

**Summary:**

This paper explores applying V:N:M sparsity to the weight parameters of transformer-based models, which can well retain accuracy while harvest inference speedup from sparse tensor cores in advanced GPUs. To achieve this, the authors propose three techniques to convert a dense transformer into a V:N:M sparse model, including a heuristic method for selecting V and M values, V:N:M-specific channel permutation, and a three-stage LoRA training approach for LLMs. They conduct extensive experiments on mainstream transformers, including DeiT, Swin Transformer, and Llama2-7B, across multiple tasks, demonstrating that their method outperforms naive 2:4 sparsity versions. Overall, this paper could be a good empirical contributions, paving the way for V:N:M sparsity compression for Transformers.

**Strengths:**

This paper is well-written and easy to follow. It integrates multiple techniques to accelerate V:N:M-sparse transformers. While the novelty of each technique is somewhat limited, the resulting performance improvements are substantial.
For theory strengths, the paper formulates the selection of V and M value as an optimization problem, balancing accuracy with speedup constraints. To solve this efficiently, they propose a two-phase sifting process to identify the optimal (V, M) combinations; Moreover, it adapts channel permutation to V:N:M sparsity, applying it to both the input and output of weight matrices to enhance accuracy for low training budgets. Finally, to ensure stable training of V:N sparse transformers, the authors integrate a three-stage LoRA training approach, consisting of Dense LoRA, Sparse LoRA with a dynamic mask, and Sparse LoRA with a fixed mask.

**Weaknesses:**

1. While it's intuitive that a smaller M results in lower sparsity in the sparse transformers, the process for excluding certain (V, M) combinations is unclear. Additionally, there is insufficient evidence in the paper demonstrating that mask diversity (MD) improves transformer accuracy.
2. In Definition 1, the objective is to maximize accuracy under speedup constraints s. However, there should be experimental evidence demonstrating that the chosen V and M values meet these constraints.
3. In Section 5.4, although V:N:M sparse Llama2 outperforms RIA and Wanda, it still suffers significant accuracy loss compared to the dense counterpart. For instance, HellaSwag accuracy drops from 57.23 to 42.88, and ARC-C from 43.3 to 34.3.
4. Equation (3) lacks the description of $\overline{M}_{t-1}$
5. In section 4.3, the definitions of $Mv$, $Wv$, $Bv$ and $ Av$ are unclear. Also, it would better to disclose detailed LoRA configurations in the three-staged LoRA training experiments (section 5.4) , such as dynamic sparse mask initialization, the interval of updating sparse masks, the actual training iteration assignment two different stages etc.
6. Figure 3 does not intuitively illustrate "in the absence of regularization, frequent updates to the masks can negatively impact the gradient flow of V:N:M sparse Transformers during fine-tuning". Additionally, adding a comprehensive empirical study would make it more convincing to draw the conclusion that "the iterations for the first two stages should not exceed 10% of the total iterations."

**Questions:**

1. For large language models (LLMs), latency is not solely determined by FLOPs; it is typically constrained by memory bandwidth and access patterns. In Definition 1, is the measured speedup based on a batch size of 1? Furthermore, when varying the batch size, will this lead to different V and M sparsity values?
2. Can channel permutation enhance the performance of SR-STE training or LoRA training?
3. How is the regularization coefficient $\lambda$ in equation (3) tuned to improve training stability?

---

> ### Author Response · Authors · 2024-11-24
> **(Q1) Grateful for your recognition and constructive feedback! Here is our answer to question 1.**
>
> We are very grateful for your recognition and constructive feedback. We aim to address the identified weaknesses and questions systematically.
>
> Q1: While it's intuitive that a smaller M results in lower sparsity in the sparse transformers, the process for excluding certain (V, M) combinations is unclear. Additionally, there is insufficient evidence in the paper demonstrating that mask diversity (MD) improves transformer accuracy.
>
> **A1**: Thank you very much for your comments.
> 1. To clarify the V and M selection process, we provide an example in Table 2.1. Assuming a speedup threshold of 2, the two-phase sifting is as follows:
>    - Coarse-Grained Sifting: For each row of V (16, 32, 64, 128) in Table 2.1, we first select the minimal M that achieves a speedup greater than 2. For instance, for V=16, M=16 is chosen, while M values of 17, 18, and 19 are excluded since their corresponding accuracy for V:N:M-sparse Transformers is lower than that with M=16, which retains more parameters. Similarly, for V=32, 64, and 128, we select M values of 16, 15, and 15, respectively. The speedups for these four (V, M) pairs—(16, 16), (32, 16), (64, 15), and (128, 15)—are highlighted in bold in Table 2.1 and exceed the threshold of 2. After coarse-grained sifting, most of the (V, M) pairs that also meet the speedup threshold are excluded, retaining only those that are critical with respect to the threshold.
>    - Fine-Grained Sifting by Mask Diversity (MD): We compute and compare the MDs of the four selected (V, M) pairs. The pair with the highest MD is selected for constructing the V:N:M-sparse Transformers.
>
> Table 2.1 A example illustrating the V and M selection process
> | Speedup | ... |   M=15   |   M=16   | M=17 | M=18 | M=19 | ... |
> |:-------:|:---:|:--------:|:--------:|:----:|:----:|:----:|:---:|
> |   V=16  | ... |    1.86   |  **2.1** | 2.15 |  2.2 | 2.24 | ... |
> |   V=32  | ... | 1.97 |   **2.16**   | 2.21 |  2.3 | 2.39 | ... |
> |   V=64  | ... |    **2.04**  | 2.21 | 2.32 |  2.4 | 2.53 | ... |
> |  V=128  | ... | **2.18** |   2.29   | 2.36 | 2.47 |  2.6 | ... |
>
> 2. To further demonstrate that MD, rather than model parameter counts, serves as a better indicator for V:N:M-sparse Transformers, we present a specific example in Table 2.2. Among three (V, M) configurations, the 128:2:15-sparse DeiT-base has the highest parameter count, yet the 16:2:16-sparse DeiT-base, which exhibits the highest MD, achieves the best accuracy, significantly surpassing the other two configurations. The principle behind the example is that, **both the sparse granularity determined by V and the retained parameter counts dictated by M influence the accuracy of V:N:M-sparse Transformers**. Compared to relying solely on parameter counts, MD accounts for both factors, thereby providing a more accurate measure of performance.
>
> Table 2.2 V:N:M-sparse DeiT-Base 30 epoch LoRA
>    accuracy on downstream tasks
>
> |       (V,M)       |  (16,16) | (32,16) | (128,15) |
> |:-----------------:|:--------:|:-------:|:--------:|
> |     Params.(M)    |   10.8   |   10.8  | **11.5** |
> |   Simplified MD   |   20837  |  18674  |   18168  |
> |   Bird Accu.(%)   |   84.1   |   82.1  |   79.5   |
> | Vehicle Accu.(%)  |   77.4   |   75.2  |   72.6   |
> | CIFAR10 Accu.(%)  |   86.5   |   85.4  |   84.0   |
> | CIFAR100 Accu.(%) |   55.9   |   53.3  |   50.5   |
> |      AVG (%)      | **76.0** |    74   |   71.7   |

---

> ### Author Response · Authors · 2024-11-24
> **(Q2, Q3, Q4) Grateful for your recognition and constructive feedback! Here are our answers to question 2, 3, 4.**
>
> Q2: In Definition 1, the objective is to maximize accuracy under speedup constraints s. However, there should be experimental evidence demonstrating that the chosen V and M values meet these constraints.
>
> **A2**: Thank you.
> 1. Using Llama2-7B as an example, we have listed the selected V and M values for different batch sizes, along with the corresponding speedups of the V:N:M-sparse Llama2-7B compared to their dense counterpart. This information is presented in Table 2.5 (located in the context of part **(Q7)**). We randomly generated multiple speedup constraints s from the interval [1, 2.5], resulting in s∈{1.14, 1.26, 1.34, 1.52, 1.65, 1.88, 2.12}. From the table, it is evident that our proposed V and M selection method yields (V, M) pairs that consistently achieve speedups exceeding the corresponding constraints. For instance, with a batch size of 8 and a speedup constraint of 1.52, the selected 128:2:6-sparse Llama2-7B achieves a speedup of 1.55, which is greater than 1.52.
> 2. Additionally, in figures such as Figure 4 and Figure 10 in our paper, the x-axis of the blue line dots represents the speedup constraints, while the x-axis of the yellow line dots represents the speedups achieved by the selected (V, M) pairs. It is clear that the yellow lines consistently lie above and to the right of the corresponding blue lines, indicating that our chosen V and M values satisfy the speedup constraints.
> 3. Finally, we further provide the V and M selection results for ViT-large, ViT-huge, and Llama2-13B for your reference, as shown in Figures 2.1, 2.2, and 2.3, respectively. Our V and M selection method performs effectively, and the selected values meet the speedup constraints across all three models as well.
>
> [Figure 2.1 Average Top-1 accuracy of ViT-large on downstream tasks with different V and M vlaues](https://anonymous.4open.science/r/conf_dat_Anonymous-8055/downstream_vit-large.pdf)
>
> [Figure 2.2 Average Top-1 accuracy of ViT-huge on downstream tasks with different V and M vlaues](https://anonymous.4open.science/r/conf_dat_Anonymous-8055/downstream_vit-huge.pdf)
>
> [Figure 2.3 Average scores of Llama2-13B on 5-shot tasks with different V and M values.](https://anonymous.4open.science/r/conf_dat_Anonymous-8055/downstream_Llama2-13B.pdf)
>
> ---
> Q3: In Section 5.4, although V:N:M sparse Llama2 outperforms RIA and Wanda, it still suffers significant accuracy loss compared to the dense counterpart. For instance, HellaSwag accuracy drops from 57.23 to 42.88, and ARC-C from 43.3 to 34.3.
>
> **A3**: Thank you for your feedback. The accuracy drop compared to the dense counterpart is primarily due to limited computational resources. In our study regarding Llama2, we conduct experiments with at most 12 **million** tokens. This amount is far from sufficient to restore accuracy to the levels achieved by the dense model. Despite this limitation, our results still notably surpass those of related works on post-training 2:4 sparsity in the field.
>
> However, we would like to highlight that given adequate training, sparse large language models (LLMs) can achieve performance levels comparable with dense counterparts, while still benefiting from the substantial speed advantages associated with sparsity. For instance, [1] utilized 7.5 **billion** tokens for continual training of the 2:4-sparse Llama2-7B, achieving scores of 55.24 on HellaSwag and 41.11 on ARC-C. Furthermore, [2] reported that with **145 billion** tokens and sparse pretraining, which setup is typically prohibitively costly for most researchers, the 70%-sparse Llama2-7B can fully recover accuracy for fine-tuning tasks.
>
> [1]. Pruning large language models with semi-structural adaptive sparse training. ARXIV2024.
>
> [2]. Enabling high-sparsity foundamental Llama models with efficient pretraining and deployment. ARXIV2024.
>
> ---
> Q4: Equation (3) lacks the description of $\overline{M}_{t-1}$.
>
> **A4**: Many thanks. In the newly uploaded version of our paper, we have included the explanation of $\overline{M}{t-1}$, clarifying that it represents the logical negation of the mask $M{t-1}$ at training step $t-1$. This mask identifies the weights that have been pruned, allowing the regularization term to focus exclusively on these weights, thereby facilitating a gradual reduction in their norms. Consequently, after sufficient training, the norms of the pruned weights in a weight matrix approach zeros, resulting in the matrix conforming to the desired sparse pattern. Additionally, we have thoroughly revised the paper to ensure that all notations in the formulas are accompanied by appropriate descriptions.

---

> ### Author Response · Authors · 2024-11-24
> **(Q5) Grateful for your recognition and constructive feedback! Here is our answer to question 5.**
>
> Q5: In section 4.3, the definitions of $Mv$, $Wv$, $Bv$, and $Av$ are unclear. Also, it would better to disclose detailed LoRA configurations in the three-staged LoRA training experiments (section 5.4) , such as dynamic sparse mask initialization, the interval of updating sparse masks, the actual training iteration assignment two different stages etc.
>
> **A5**: Thanks. The $Mv$, $Wv$, $Bv$, and $Av$ are typos and should be corrected as $M$, $W$, $B$, and $A$, respectively. We have made these revisions throughout the paper to ensure consistency and clarity. We apologize for the oversight and sincerely appreciate your carefulness.
>
> Besides, we are very pleased to detail the configurations used in our three-stage LoRA training as outlined in Table 2.3:
>
> Table 2.3 the details of our three-staged LoRA training
> | Items                                | Settings                                                                 |
> |--------------------------------------|--------------------------------------------------------------------------|
> | The initial of dynamic sparse masks  | RIA-based pruning                                                        |
> | Interval of updating sparse masks    | 20 iterations per update                                                 |
> | Actual training iteration assignment | (A total of 12000) First stage: 320; Second stage: 320; Third stage: 11360 |
> | Overall LoRA training tokens         | 1024 tokens $\times$ 12000 iterations = 12 million tokens                |
> | LoRA rank                            | 16                                                                       |
> | LoRA $\alpha$                        | 32                                                                       |
> | LoRA modules in Llama2               | q_porj, k_proj, v_proj, o_proj, up_proj, gate_proj, down_proj            |
>
> Specifically,
> - To obtain the initial dynamic sparse masks $M$ for the weight matrix $W$ in the second stage of LoRA training, we first merge the LoRA matrices $BA$ with $W$. RIA-based pruning is then applied to the merged matrix, where the retained weights are accordingly assigned a value of 1 in $M$, while the pruned weights are assigned a value of 0 in $M$.
> - We set the interval for updating sparse masks to 20 iterations. A smaller interval can destabilize LoRA training, a phenomenon also noted in DeiT-base in our paper (Please refer to Table 1 in our paper).
> - We adjust hyperparameters, including the mask update intervals and update counts, to ensure that the first, second, and third stages account for approximately 2.5%, 2.5%, and 95% of the total training, respectively.
> - Following standard practice, we use 1,024 tokens for each training iteration, resulting in a total of 12 million tokens for our LoRA training.
> - The ranks of the LoRA matrices, specifically $A$ and $B$, are set to 16, with LoRA $\alpha$ configured to 32 to maintain the regular setting of $\alpha/\text{rank} = 2$.
> - All the linear layers in Llama2 are equipped with LoRA training.

---

> ### Author Response · Authors · 2024-11-24
> **(Q6) Grateful for your recognition and constructive feedback! Here is our answer to question 6.**
>
> Q6: Figure 3 does not intuitively illustrate "in the absence of regularization, frequent updates to the masks can negatively impact the gradient flow of V:N:M sparse Transformers during fine-tuning". Additionally, adding a comprehensive empirical study would make it more convincing to draw the conclusion that "the iterations for the first two stages should not exceed 10% of the total iterations."
>
> **A6**: Thanks.
> 1. We would like to clarify that gradient flow, in our context, specifically refers to the change in gradient norms during the LoRA training process. As shown in Figure 3 in the main paper, the gradient norms for LoRA with dynamic masks are consistently lower than those with fixed masks. Such low gradient norms may lead to underfitting in neural networks, undermining the gains achieved through mask exploration and negatively impacting accuracy [3].
> 2. To further intuitively illustrate the necessity for infrequent mask updates, we present the loss change curves for sparse LoRA fine-tuning with both fixed and dynamic masks. As depicted in Figure 2.4 below, constant mask updates result in a progressively higher training loss compared to the fixed-mask training as the training progresses.
>
>  [Figure 2.4 Loss changes of Llama2-7B during sparse LoRA fine-tuning with dynamic
> masks and fixed masks, respectively.](https://anonymous.4open.science/r/conf_dat_Anonymous-8055/lora_loss_change.pdf)
>
> [3] Training your sparse neural network better with any mask. ICML2022.
>
> 3. We conducted experiments with Llama2-7B to assess the impact of the proportion of the first two stages relative to total iterations. As shown in Table 2.4, a proportion of 5% yields the best results for both 64:2:5 and 64:2:6 configurations. Moreover, when the proportions exceed 10%, the results become unstable.
>
> Table 2.4 5-shot average scores for the different proportions of the first two stages in Llama2-7B LoRA training
> |   64:2:5   | 64:2:5 |   64:2:6   | 64:2:6 |
> |:----------:|:------:|:----------:|:------:|
> | Proportion |  Score | Proportion |  Score |
> |     5%     |  **51.75** |     5%     |  **46.03** |
> |     10%    |  50.94 |     10%    |  45.75 |
> |     15%    |  51.11 |     15%    |  44.24 |
> |     20%    |  49.99 |     20%    |  45.23 |

---

> ### Author Response · Authors · 2024-11-24
> **(Q7) Grateful for your recognition and constructive feedback! Here is our answer to question 7.**
>
> Q7: For large language models (LLMs), latency is not solely determined by FLOPs; it is typically constrained by memory bandwidth and access patterns. In Definition 1, is the measured speedup based on a batch size of 1? Furthermore, when varying the batch size, will this lead to different V and M sparsity values?
>
> **A7**: Thanks.
> 1. The speedup results presented in our paper are based on a batch size of 1.
> 2. When varying the batch size, the selected values for V and M differ accordingly. For example, the results for Llama2-7B at various batch sizes are shown in Table 2.5, where the selected V and M values are highlighted in bold for clarity. The speedup thresholds are randomly generated from the range [1, 2.5]. Since the speedup of a V:N:M-sparse Transformer compared to its dense counterpart varies with different batch sizes, the selection results will also differ. We consider this phenomenon to be normal and it does not impact our technical contributions. In practical LLM inference systems, especially in cloud environments, multiple queries are often concatenated into a fixed batch before inference, which scenario is still suitable for the application of our techniques.
>
> Table 2.5 Selected V and M values of Llama2-7B under different batch sizes (BS). **(X,4)** means 2:4 sparsity
> |   　  |     Speedup threshold     |     1.14    |     1.26     |     1.34     |     1.52     |     1.65     |      1.88     |      2.12     |
> |:-----:|:-------------------------:|:-----------:|:------------:|:------------:|:------------:|:------------:|:-------------:|:-------------:|
> |  bs=1 |     **Selected (V,M)**    |  **(X, 4)** |  **(X, 4)**  |  **(64, 5)** | **(128, 5)** | **(128, 5)** |  **(128, 7)** |  **(128, 8)** |
> |  bs=1 | Speedup of selected (V,M) |     1.26    |     1.26     |     1.49     |     1.65     |     1.65     |      1.99     |      2.16     |
> |  bs=2 |     **Selected (V,M)**    |  **(X, 4)** |  **(64, 5)** |  **(64, 5)** | **(128, 5)** | **(128, 6)** |  **(128, 8)** | **(128, 10)** |
> |  bs=2 | Speedup of selected (V,M) |     1.19    |     1.36     |     1.36     |     1.54     |      1.7     |      2.01     |      2.17     |
> |  bs=4 |     **Selected (V,M)**    | **(64, 5)** |  **(64, 5)** | **(128, 5)** | **(128, 6)** | **(128, 7)** |  **(128, 8)** | **(128, 11)** |
> |  bs=4 | Speedup of selected (V,M) |     1.26    |     1.26     |     1.45     |      1.6     |     1.74     |      1.88     |      2.12     |
> |  bs=8 |     **Selected (V,M)**    | **(64, 5)** | **(128, 5)** | **(128, 5)** | **(128, 6)** | **(128, 7)** |  **(128, 9)** | **(128, 13)** |
> |  bs=8 | Speedup of selected (V,M) |     1.14    |     1.34     |     1.34     |     1.55     |     1.68     |      1.88     |      2.14     |
> | bs=16 |     **Selected (V,M)**    | **(64, 5)** | **(128, 5)** | **(128, 6)** | **(128, 7)** | **(128, 8)** | **(128, 11)** | **(128, 13)** |
> | bs=16 | Speedup of selected (V,M) |     1.14    |     1.29     |     1.41     |     1.55     |     1.72     |      1.91     |    2.12       |

---

> ### Author Response · Authors · 2024-11-24
> **(Q8, Q9) Grateful for your recognition and constructive feedback! Here are our answers to question 8, 9.**
>
> Q8: Can channel permutation enhance the performance of SR-STE training or LoRA training?
>
> **A8**: Yes.
> 1. Our V:N:M-specific channel permutation (CP) does enhance the LoRA training performance. For instance, as shown in Table 3 in the main text of our paper, our CP improves the 5-shot score of Llama2-7B under low LoRA training budgets by 1.25%. In visual tasks, our CP increases the accuracy of the 64:2:5-sparse DeiT-base during LoRA training on downstream tasks by 1.63% (Table 4 in the main text of our paper). Moreover, the improvements resulting from our CP are consistent across different V:N:M ratios, as demonstrated in Tables 2.6, 2.7, and 2.8 below.
> 2. The influence of CP on SR-STE training is not pronounced in our experiments.
>
> Table 2.6 Results of 64:2:16 DeiT-base on downstream tasks with 30 training epochs
> | Downstream tasks | BIRD | BIRD | BIRD | VEHICLE | VEHICLE | VEHICLE | CIFAR-10 | CIFAR-10 | CIFAR-10 | CIFAR-100 | CIFAR-100 | CIFAR-100 | Average |
> | --- | --- | --- | --- | --- | --- | --- | --- | --- | --- | --- | --- | --- | --- |
> | Dense model(%) | 97.7 | 97.7 | 97.7 | 97.3 | 97.3 | 97.3 | 98.1 | 98.1 | 98.1 | 87.54 | 87.54 | 87.54 | |
> | Permutation Method | No | Ours | △ | No | Ours | △ | No | Ours | △ | No | Ours | △ |  |
> | Upon Pruning(%) | 7.40 | 8.10 | 0.6 | 5.00 | 7.00 | 2.0 | 10.8 | 12.4 | 1.6 | 1.00 | 1.35 | 0.35 | **1.14** |
> | **30 epochs-LoRA**(%) | 79.7 | 84.1 | 4.3 | 74.3 | 77.4 | 3.1 | 82.9 | 86.5 | 3.6 | 50.2 | 55.9 | 5.7 | **4.18** |
>
> Table 2.7 Results of 32:2:16 DeiT-base on downstream tasks with 30 training epochs
> | Downstream tasks | BIRD | BIRD | BIRD | VEHICLE | VEHICLE | VEHICLE | CIFAR-10 | CIFAR-10 | CIFAR-10 | CIFAR-100 | CIFAR-100 | CIFAR-100 | Average |
> | --- | --- | --- | --- | --- | --- | --- | --- | --- | --- | --- | --- | --- | --- |
> | Dense model(%) | 97.7 | 97.7 | 97.7 | 97.3 | 97.3 | 97.3 | 98.1 | 98.1 | 98.1 | 87.54 | 87.54 | 87.54 |  |
> | Permutation Method | No | Ours | △ | No | Ours | △ | No | Ours | △ | No | Ours | △ |  |
> | Upon Pruning(%) | 8.10 | 6.10 | -2.0 | 6.60 | 6.20 | -0.4 | 9.98 | 13.6 | 3.6 | 1.24 | 1.21 | -0.03 | **0.29** |
> | **30 epochs-LoRA**(%) | 78.8 | 82.1 | 3.3 | 72.9 | 75.2 | 2.3 | 81.2 | 85.4 | 4.2 | 48.2 | 53.3 | 5.1 | **3.73** |
>
> Table 2.8 Results of 128:2:15 DeiT-base on downstream tasks with 30 training epochs
> | Downstream tasks | BIRD | BIRD | BIRD | VEHICLE | VEHICLE | VEHICLE | CIFAR-10 | CIFAR-10 | CIFAR-10 | CIFAR-100 | CIFAR-100 | CIFAR-100 | Average |
> | --- | --- | --- | --- | --- | --- | --- | --- | --- | --- | --- | --- | --- | --- |
> | Dense model(%) | 97.7 | 97.7 | 97.7 | 97.3 | 97.3 | 97.3 | 98.1 | 98.1 | 98.1 | 87.54 | 87.54 | 87.54 |  |
> | Permutation Method | No | Ours | △ | No | Ours | △ | No | Ours | △ | No | Ours | △ |  |
> | Upon Pruning(%) | 5.00 | 4.50 | -0.5 | 6.60 | 7.10 | 0.5 | 10.2 | 12.4 | 2.2 | 0.97 | 1.32 | 0.35 | **0.64** |
> | **30 epochs-LoRA**(%) | 77.2 | 79.5 | 2.3 | 70.4 | 72.6 | 2.2 | 79.7 | 84.0 | 4.3 | 47.0 | 50.5 | 3.5 | **3.08** |
>
> ---
> Q9: How is the regularization coefficient $\lambda$ in equation (3) tuned to improve training stability?
>
> **A9**:  Thank you very much. We utilize the method described in [4] to tune $\lambda$. Specifically, we first perform a grid search to compute the flip rate for each candidate $\lambda$. Subsequently, we select a $\lambda$ with a "healthy" flip rate as defined in [4] for our training process.
>
>
> [4]: Accelerating Transformer Pre-training with 2:4 Sparsity. ICML2024.

---

> > ### Comment · Reviewer_W4Dx · 2024-11-27
> > **Post-rebuttal comments**
> >
> > The authors have addressed my most concerns and I will keep my score.

---

### Official Review · Reviewer_apM6 · 2024-11-04

**Soundness:** 3
**Presentation:** 3
**Contribution:** 3
**Rating:** 6
**Confidence:** 3

**Summary:**

This paper investigates the use of V:N:M sparsity in Transformer models and proposes three key methods to enhance its performance: heuristic V and M selection, V:N:M channel permutation, and a three-stage LoRA training technique. The experimental results demonstrate that V:N:M sparsity offers a broader range of speedup-accuracy trade-offs compared to traditional 2:4 sparsity.

**Strengths:**

1. This paper introduces a comprehensive framework that generates highly accurate V:N:M-sparse Transformers under different constraints, allowing users to balance the system performance and model accuracy.

2. The paper addresses key challenges in applying V:N:M sparsity by introducing three innovative techniques: heuristic V and M selection, V:N:M channel permutation, and a three-stage LoRA training technique.

3. Experiments show superior performance of proposed methods.

**Weaknesses:**

1. The paper lacks a comparison with the latest related works, which limits its ability to contextualize the advantages and innovations within the current research landscape, such as [1]. Moreover, please also report the model performance without sparsity in Figure 4,5,7,8,9 to show the influence of introducing model sparsity.

2. The experiments in this paper are limited to small-scale models, such as ViT, DeiT, and Llama-7B. It does not extensively explore the performance on larger models, like ViT-Huge or LLaMA models with higher parameter counts, where the effectiveness of the sparsity method might vary.

3. The work would be stronger if it provided performance benchmarks for Llama2-7B on pretraining benchmarks, such as MMLU or GSM-8k, to give a more comprehensive evaluation of the model's capabilities across a broader set of important benchmarks. If computational resources are limited, consider performing post-training on Llama2-7B to demonstrate the applicability of this paper's methods.

[1] Yang H, Yin H, Shen M, et al. Global vision transformer pruning with hessian-aware saliency[C]//Proceedings of the IEEE/CVF conference on computer vision and pattern recognition. 2023: 18547-18557.

**Questions:**

This paper propose three techniques, including heuristic V and M selection, V:N:M channel permutation, and a three-stage LoRA training technique. Please further explain the relationship between these techniques, are they complementary, sequential, or something else?

---

> ### Author Response · Authors · 2024-11-24
> **(Q1, Q2) Grateful for your recognition and helpful suggestions! Here are our answers to question 1, 2.**
>
> Q1: The paper lacks a comparison with the latest related works, which limits its ability to contextualize the advantages and innovations within the current research landscape, such as [1]. Moreover, please also report the model performance without sparsity in Figure 4,5,7,8,9 to show the influence of introducing model sparsity.
>
> **A1**: Thank you very much for your constructive feedback.
>
> 1. We have included [1], referred to as NViT, in Table 5 of our newly revised paper as one of the related works for comparison. Here, we provide further details on the comparison: 1) In terms of accuracy, both our method and NViT achieve nearly lossless Top-1 accuracy for sparse DeiT-base and DeiT-small. 2) Regarding FLOPs reduction, our method achieves the highest reduction for both DeiT-base (71.6% ↓) and DeiT-small (54.3% ↓) when compared with NViT and other related works. 3) For parameter reduction, our method achieves the highest reduction for DeiT-small (57.9% ↓) among related works, while NViT achieves the highest reduction for DeiT-base (80.4% ↓), compared to our 73.8% ↓. It is noteworthy that NViT combines multiple strategies for compressing DeiTs, including global structural pruning, 2:4 pruning, and parameter redistribution. In contrast, our work focuses solely on V:N:M sparsity and has the potential to further enhance parameter reduction ratios when combined with other compression strategies. 4) For speedups, both NViT and our method yield significant practical speedups for sparse DeiTs. Specifically, NViT-B with ASP achieves speedups of 1.86x and 1.85x on V100 and RTX 3080 GPUs, respectively, while our 64:2:8-sparse DeiT-base achieves a 2.08x speedup on RTX 3090 GPUs.
> 2. We have added the performance results without sparsity in Figures 4, 5, 7, 8, and 9 in our revised paper. Particularly for Llama2-7B, we present results for both the dense model and the 2:4-sparse model; the latter serves as the primary benchmark for comparing the performance of our V:N:M-sparse Llama2. Please note that Figure 7 has been re-numbered as Figure 10 in the latest version of our paper.
> ___
> Q2: The experiments in this paper are limited to small-scale models, such as ViT, DeiT, and Llama-7B. It does not extensively explore the performance on larger models, like ViT-Huge or LLaMA models with higher parameter counts, where the effectiveness of the sparsity method might vary.
>
> **A2**: Thank you for your concern. We endeavor to provide four results on larger models:
> 1. For LLM LoRA training, we present the average scores of **Llama2-13B** on 5-shot tasks with different V and M values, as depicted in [Figure 1.1](https://anonymous.4open.science/r/conf_dat_Anonymous-8055/downstream_Llama2-13B.pdf). The average score for the dense model is 68.23, while the average score for the 2:4-sparse model is 56.78.
> 2. For visual downstream tasks, we report the average Top-1 accuracy for ViT-large and **ViT-huge** with different V and M values. These are illustrated in [Figure 1.2](https://anonymous.4open.science/r/conf_dat_Anonymous-8055/downstream_vit-large.pdf) and [Figure 1.3](https://anonymous.4open.science/r/conf_dat_Anonymous-8055/downstream_vit-huge.pdf), respectively. The average Top-1 accuracy for the dense ViT-large is 94.53%, while for the dense ViT-huge, it is 92.99%.
>
> Figures 1.1, 1.2, and 1.3 demonstrate that Training Setting 1 (TS1) and Training Setting 3 (TS3) in our proposed framework, along with the novel techniques integrated within it, perform effectively for larger models.
>
> 3. Regarding approximate visual pretraining, we benchmark the accuracy of the 64:2:8-sparse ViT-large trained on ImageNet1K, as shown in Table 1.1 below. The 64:2:8-sparse ViT-large does not fully recover the accuracy of its dense counterpart due to non-strict pretraining pipelines. Specifically, the dense ViT-large is pretrained on ImageNet21k using TPUv3 hardware and subsequently fine-tuned on ImageNet1K, achieving a Top-1 accuracy of 84.31%. However, pretraining the 64:2:8-sparse ViT-large on ImageNet21k is challenging with our limited computational resources and infrastructure. Nevertheless, our 64:2:8-sparse ViT-large achieves a Top-1 accuracy of 81.24% when only training on ImageNet1K with a speedup of 1.88x compared to the dense model.
>
> Table 1.1 Accuracy of 64:2:8-sparse ViT-large on ImageNet1K
> | ViT-large     | Top-1 acc. (%) | Top-5 acc. (%) | Speedup |
> |---------------|----------------|----------------|---------|
> | Dense         | 84.31          | 97.19          | 1       |
> | 64:2:8-sparse | 81.24          | 95.34          | 1.88    |

---

> ### Author Response · Authors · 2024-11-24
> **(Q3) Grateful for your recognition and helpful suggestions! Here are our answers to question  3.**
>
> Q3: The work would be stronger if it provided performance benchmarks for Llama2-7B on pretraining benchmarks, such as MMLU or GSM-8k, to give a more comprehensive evaluation of the model's capabilities across a broader set of important benchmarks. If computational resources are limited, consider performing post-training on Llama2-7B to demonstrate the applicability of this paper's methods.
>
> **A3**: Thank you very much for your suggestion. As you noted, GSM8K and MMLU serve primarily as pretraining benchmarks, actually requiring extensive pretraining or continuous training of sparse LLMs with billions of tokens prior to evaluation. In contrast, our three-staged LoRA training method is specifically designed for post-pretraining fine-tuning using millions of tokens. Nevertheless, we sincerely appreciate your suggestion and evaluated the performance of 64:2:5-sparse Llama-7B on the GSM-8K benchmark by post-pretraining fine-tuning using our proposed method. The results are presented in Table 1.2 below.
>
> | Llama2-7B    | Score | Speedup |
> |--------------|-------|---------|
> | Dense        | 30.93 | 1       |
> | Wanda(2:4)   | 7.58  | 1.26    |
> | Ours(64:2:5) | 17.13 | 1.49    |
>
> The results indicate that the mathematical capabilities of Llama2-7B are significantly compromised by sparsity, both in the 2:4 and V:N:M configurations. However, our 64:2:5-sparse Llama2-7B still outperforms the post-trained 2:4-sparse counterpart. Notably, even after continual training with 7.5 **billion** tokens, the 2:4-sparse Llama2-7B still exhibits a score that is over 4 points lower than that of the dense model on GSM-8K [2]. It is important to highlight that our LoRA training for the sparse Llama2-7B utilized only 12 **million** tokens.
>
> Regarding the MMLU benchmark, both the 2:4-sparse and 64:2:5-sparse Llama2-7B scores are close to random guessing, at approximately 25%, following LoRA training only with 12 million tokens. Generally, for knowledge-intensive tasks, N:M sparsity, including V:N:M sparsity, demonstrates ineffectiveness with limited tokens [3]. We consider this an area for future research.
>
> [2]. Pruning large language models with semi-structural adaptive sparse training. ARXIV2024.
>
> [3]. Compressing LLMs: the truth is rarely pure and never simple. ICLR2024.

---

> ### Comment · Reviewer_apM6 · 2024-11-26
> **Response to A1 & A2**
>
> Thanks for your response.
>
> 1. Why does NViT-B(ASP) achieve better Top-1 accuracy than the original dense model in Table 5?
> + For instance, NViT-B(ASP) reports a Top-1 accuracy of 83.29 for DeiT-B (compared to the dense model's 81.84) and 82.19 for DeiT-S (compared to the dense model's 79.85).
> + Could you provide further explanations for this result? Were the baseline results for the dense models taken directly from other papers or re-evaluated in your experiments?
>
> 2. The reported accuracy for Llama-7B-dense is 61.99 (Figure 4), while Llama-13B-dense achieves 68.23 (Figure 8). Given that Llama-7B-dense has approximately 2x the inference efficiency of Llama-13B-dense, the V:N:M sparse models in Figure 8 exhibit lower accuracy than Llama-7B-dense, with smaller speedup gains compared to 7B. Does this imply that the V:N:M sparsity approach is less/not effective for Llama models?
>
> 3. Based on current results, the V:N:M sparsity approach demonstrates better suitability for DeiT and ViT models, where the performance trade-offs are more favorable.

---

> ### Comment · Reviewer_apM6 · 2024-11-26
> **Response to A3**
>
> Thanks for your response.
>
> Based on the post-training results for the Llama models, it seems that V:N:M sparsity is not well-suited for Llama model training. While it does show improvements compared to the baseline, the accuracy loss can not be neglected relative to the 1.5x speedup it provides.
>
> I will maintain my current score, but I encourage you to continue optimizing your work further. Best of luck!

---

> ### Author Response · Authors · 2024-11-27
> **Grateful for your reply!**
>
> Thank you very much for your concern. We appreciate the opportunity to provide further explanation.
>
> **Explanation 1**: The improved Top-1 accuracy of NViT-B(ASP) and NViT-S(ASP) compared to the original dense model, is due to **knowledge distillation (KD), which is explicitly acknowledged in the NViT paper**.
> - For a fair comparison, NViT-B(ASP) and NViT-S(ASP) should be compared to the dense model also under the same KD conditions to derive the accuracy difference (represented as $\Delta$ in our paper). For instance, the accuracy of the dense DeiT-base models under KD is 83.6%. In Table 5, We exactly use the accuracy of the KD-based dense models to calculate $\Delta$. Besides, in our paper, $\Delta$ for NViT is marked with an asterisk to indicate its specific computation method, detailed in Appendix J.
> - It is important to note that KD is generally considered parallel to other model compression methodologies, such as sparsification (including V:N:M sparsity) and quantization. Considering this, we use the results of the pure dense model without KD as our baseline, allowing for a clearer evaluation of the individual efficacy of our V:N:M sparsification method. This manner of using a pure dense model without KD as a baseline is more widely adopted in the literature.
> - The baseline results without KD are identical to those reported in the original DeiT paper [4] and have been accurately reproduced in our experiments.
>
> [4] Training data-efficient image transformers & distillation through attention. ICML2021.
>
> **Explanation 2**: The absolute accuracy of sparse Llama models, including Llama2-7B and Llama2-13B, can be significantly enhanced on benchmarks such as question-answering tasks and GSM8K through increased training token counts coupled with adequate training, such as pretraining and continual training.
>
> - In addition to the previously mentioned study [2], which employs **7.5 billion** tokens for continual training of sparse Llama2-7B to narrow the accuracy gap to 4.1% compared to its dense counterpart on GSM8K tasks, another study [5] utilizes **145 billion** tokens to pretrain a sparse Llama2 backbone, achieving full recovery of fine-tuning accuracy for tasks including GSM8K. These findings suggest the considerable potential of sparse LLMs. Meanwhile, this level of resource requirements is typically prohibitive for general research.
>
> - We would like to clarify that our three-staged LoRA training method is specifically designed for post-pretraining fine-tuning using **millions** of tokens, which for the first time enables V:N:M-sparse LLMs to outperform their post-training 2:4 counterparts in a highly data-efficient manner. We kindly deem that optimizing **absolute accuracy on pretraining or continual training-dependent benchmarks** like GSM8K and MMLU is beyond the scope of our current work and is better suited for future research.
>
> Thank you so much for your empathetic understanding and valuable reviews!
>
> [5]. Enabling high-sparsity foundamental Llama models with efficient pretraining and deployment. ARXIV2024.

---

### Comment · Area_Chair_psTF · 2024-11-23
**Engage in Discussions Before Nov 26 (AoE)**

Dear Reviewers,

First, let me thank you for your invaluable contributions to the ICLR review process. Your constructive feedback plays a key role in enhancing the quality of submissions.

---

As we approach the final days of the discussion phase (ending **Nov 26, 2024, AoE**), I kindly remind you to:

- Please take a moment to review the authors' responses to your comments. This is an opportunity to clarify any remaining questions, acknowledge misunderstandings, and refine your evaluation.

- If you need further clarification, don't hesitate to post your comments as soon as possible.

- If the authors' responses address your concerns or provide new insights, please consider updating your score to reflect this.

---

Your thoughtful participation during this phase is especially valuable for borderline papers, where additional input can be critical to ensuring a fair decision-making process.

I understand how busy this time of year can be and truly appreciate the time and care you dedicate to this important role. Your efforts make a tangible impact on the success of ICLR.

Thank you once again for your dedication.

Best regards,

Area Chair, ICLR 2025

---

### Author Response · Authors · 2024-12-03
**Summary of Rebuttal (Many Thanks to All Reviewers, Area Chairs, and Program Chairs)**

We sincerely thank all reviewers for their thoughtful evaluations. We are encouraged by their recognition of our clear and well-organized motivation (QK6w), comprehensive evaluations across both vision Transformers and LLMs (apM6, WKMR, QK6w, Dgv5), substantial performance improvements (W4Dx), and the property of "easy to follow" (W4Dx, Dgv5, QK6w). Below, we briefly re-emphasize some key points:

## Distinction from Initial Work
The initial work for V:N:M sparsity focused on GPU kernel acceleration, while our research centers on algorithmic design aimed at enhancing the accuracy and applicability of V:N:M-sparse Transformers. Our work and the initial study operate at different levels of V:N:M sparsity.

## Novelty of Our Work
We introduce three innovative methods that significantly improve the accuracy of various V:N:M-sparse Transformers under different constraints: **a heuristic V and M selection method**, **a V:N:M-specific channel permutation (CP) method**, and **a three-staged LoRA training method**. Each of these methods is supported by our insights into V:N:M sparsity and addresses challenges that have never before been explored in this domain.

## Key Insights in Our Work
1. We demonstrate that, contrary to common assumptions, higher parameter counts do not necessarily correlate with improved accuracy for V:N:M-sparse Transformers. Both the sparse granularity defined by V and the retained parameter counts dictated by M significantly impact accuracy.
2. We reveals that CP for V:N:M sparsity differs fundamentally from that used for 2:4 sparsity, as both input and output channel permutations affect retained parameter norms for V:N:M-sparse weight matrices.
3. We empirically prove for the first time that brief dynamic mask changes following dense LoRA initialization can greatly enhance LoRA training effectiveness.

## Contributions of Our Work
1. Our proposed framework within the three novel methods allows V:N:M-sparse Transformers to achieve comparable or superior accuracy relative to 2:4 sparsity. Notably, with our efforts, the DeiT-small and DeiT-base models maintain lossless accuracy at 64:2:5 and 64:2:8 sparsity, respectively. Given that V:N:M-sparse Transformers achieve higher speedups and offer a broader spectrum of speedup-accuracy trade-offs than 2:4 sparsity, our research promotes the widespread implementation of V:N:M sparsity as an effective strategy for Transformer compression. **We establish that V:N:M sparsity outperforms 2:4 sparsity in compressing highly redundant Transformers**.
2. More broadly, our proposed three methods are applicable to neural architectures where linear layers play predominant roles, such as ResMLP.

Additionally, we supplement extensive experiments during the rebuttal phase, including the following (note that the figure and table numbers refer to those in the rebuttal webpage for your convenience):
- Results for large-scale V:N:M-sparse Transformers (ViT-large in [Figure 3.1](https://anonymous.4open.science/r/conf_dat_Anonymous-8055/downstream_vit-large.pdf), ViT-huge in [Figure 3.2](https://anonymous.4open.science/r/conf_dat_Anonymous-8055/downstream_vit-huge.pdf), and Llama2-13B in [Figure 3.3](https://anonymous.4open.science/r/conf_dat_Anonymous-8055/downstream_Llama2-13B.pdf)).
- V and M selection results across various batch sizes (Table 2.5).
- An ablation study on the proportion of the first two stages in our three-staged LoRA training method (Table 2.4).
- Comparisons of loss changes between fixed and dynamic LoRA training ([Figure 2.4](https://anonymous.4open.science/r/conf_dat_Anonymous-8055/lora_loss_change.pdf)).
- The impact of CP on LoRA training (Tables 2.6, 2.7, and 2.8).
- Results showing V:N:M-sparse Transformers with more parameters but lower accuracy (Table 3.5).
- Results for V:N:M-sparse ResMLP with our three proposed methods ([Figure 4.1](https://anonymous.4open.science/r/conf_dat_Anonymous-8055/downstream_resmlp-b24.pdf), Table 4.3).

Moreover, we provide clarifications on the following aspects:
- Details on the motivation and principles of mask diversity (A1 to W4Dx, A4 to WKMR).
- Evidence that selected V and M values meet optimization objectives (A2 and A7 to W4Dx).
- Hyperparameters used in the three-staged LoRA training method (A5 to W4Dx).
- The distinctions between our work and the initial study regarding V:N:M sparsity (A4 to WKMR, Part 1 to Dgv5).
- Specific GPU cores employed to accelerate different sparse patterns (A1 to QK6w).
- The importance of training token counts and resources for achieving high absolute accuracy in sparse LLMs (Explanation 2 to apM6, A3 to W4Dx).

All significant experiments and clarifications have been integrated and highlighted in our revised paper, which also addresses several typographical errors.

Best wishes,

authors

---

### Meta-Review · Area_Chair_psTF · 2024-12-25

**Metareview:**

This paper explores the use of V:N:M sparsity to accelerate ML models, addressing limitations of the conventional 2:4 sparsity format. The authors propose three techniques to enhance the accuracy and applicability of V:N:M sparsity pattern: (a) a heuristic to select V and M values to balance between accuracy and speedup, (b) a V:N:M-specific channel permutation technique to improve performance under limited training budgets, (c) a three-staged LoRA training to maintain accuracy during fine-tuning.

## Strengths

- The proposed framework applies to different architectures, including both vision models and LLMs, which broadens its potential impact (Reviewer WKMR, Reviewer QK6w).
- The introduction of V and M selection using mask diversity (MD) contributes to the understanding of accuracy-speedup trade-offs (Reviewer WKMR).
- The channel permutation method and three-staged LoRA training approach add practical value for fine-tuning sparse models under limited resources (Reviewer W4Dx, Reviewer QK6w).
- The authors run extensive evaluations on various benchmarks, demonstrating the potential of V:N:M sparsity to achieve competitive results in different scenarios (Reviewer QK6w, Reviewer W4Dx).


## Weaknesses

- V:N:M sparsity underperforms significantly in Llama models, as highlighted by Reviewer apM6. Despite achieving a 1.5x speedup, the accuracy drop relative to the dense baseline cannot be overlooked. This suggests that V:N:M sparsity may not generalize well to LLMs.
- Reviewer apM6 specifically noted:

> Based on the post-training results for the Llama models, it seems that V:N:M sparsity is not well-suited for Llama model training. While it does show improvements compared to the baseline, the accuracy loss cannot be neglected relative to the 1.5x speedup it provides.

- Several reviewers (Dgv5, WKMR) questioned the originality of the contributions, noting that the work appears to build on existing studies like Venom without sufficiently distinguishing itself. Reviewer Dgv5 stated:

> I think there are still 'lack of novelty' of this paper.

- The heuristic nature of V and M selection raises doubts about its scalability to larger models, as pointed out by Reviewer WKMR. Furthermore, the experiments do not adequately validate whether the proposed methods consistently meet the claimed speedup and accuracy trade-offs under different configurations.

- The work heavily relies on the Venom GPU kernel for speedup, which detracts from the technical contributions of the paper (Reviewer Dgv5, Reviewer QK6w).

---

The primary reason for recommending **rejection** is the inconsistent results and limited generalizability of V:N:M sparsity, particularly for LLMs. As Reviewer apM6 noted, the accuracy loss for Llama models is substantial and cannot be justified by the modest speedup achieved. This limitation directly challenges the claim of broad applicability and undermines the practical impact of the proposed framework.

While the paper introduces some promising ideas, the inconsistent results and lack of sufficient contributions convinced me to recommend rejection. Future work could focus on addressing the limitations in LLMs, improving the scalability of the techniques, and establishing clearer distinctions from prior work.

**Additional Comments On Reviewer Discussion:**

The authors engaged actively with reviewers, addressing several concerns through clarifications, additional experiments, and updated results. However, some critical issues remained unresolved, which factored heavily into the recommendation.

- (Reviewer apM6) highlighted significant accuracy loss in Llama models relative to the dense baseline, even with a modest 1.5x speedup. This raised concerns about the generalizability of V:N:M sparsity to LLMs. The response did not adequately address the core issue of V:N:M sparsity's limited applicability to LLMs. The dependency on larger training datasets and computational resources weakens the claim of a broadly applicable framework.
- (Reviewers Dgv5 and WKMR) questioned the novelty of the paper, noting overlap with the Venom paper, which first introduced V:N:M sparsity. Reviewer Dgv5 specifically criticized the lack of distinction from prior work. While the authors provided detailed distinctions and clarified their contributions, the overlap with Venom and the incremental nature of the proposed techniques made the novelty less convincing.
- (Reviewer WKMR) raised concerns about whether the heuristic V and M selection method could scale to larger models. Reviewer QK6w also questioned its impact in certain experiments, such as Fig. 8. While the additional experiments demonstrated some scalability, concerns remained about the heuristic nature of the selection process and its robustness across diverse tasks.
- (Reviewer W4Dx) sought additional evidence on how V:N:M-specific channel permutation improved accuracy and its impact on training stability. The authors’ detailed response strengthened the paper’s contribution in this area.

---

### Decision · Program_Chairs · 2025-01-22

Reject